# Hybrid symmetry class topological insulators

## Sanjib Kumar Das and Bitan Roy[1][†]

**1** Department of Physics, Lehigh University, Bethlehem, Pennsylvania, 18015, USA

## Abstract

Traditional topological materials belong to different Altland-Zirnbauer symmetry classes (AZSCs) depending on their non-spatial symmetries. Here we introduce the notion of hybrid symmetry class topological insulators (HSCTIs): A fusion of two different AZSC topological insulators (TIs) such that they occupy orthogonal Cartesian hyperplanes and their universal massive Dirac Hamiltonian mutually anticommute, a mathematical procedure we name hybridization. The boundaries of HSCTIs can also harbor TIs, typically affiliated with an AZSC that is different from the ones for the parent two TIs. As such, a fusion or hybridization between planar class AII quantum spin Hall and vertical class BDI Su-Schrieffer-Heeger insulators gives birth to a three-dimensional class A HSCTI, accommodating quantum anomalous Hall insulators (class A) of opposite Chern numbers and quantized Hall conductivity of opposite signs on the top and bottom surfaces. Such a response is shown to be stable against weak disorder. We extend this construction to encompass crystalline HSCTI and topological superconductors (featuring half-quantized thermal Hall conductivity of opposite sings on the top and bottom surfaces), and beyond three spatial dimensions. Non-trivial responses of three-dimensional HSCTIs to crystal defects (namely edge dislocations) in terms of mid-gap bound states at zero energy around its core only on the top and bottom surfaces are presented. Possible (meta)material platforms to harness and engineer HSCTIs are discussed.

# 1   Introduction

Twenty first century physics thus far is heavily influenced by the topological classification of quantum materials [1–10]. It roots back in the discovery of quantum Hall states in the 1980s [11–13]. Over the time, topological insulators (TIs), featuring insulating bulk but gapless boundaries via a bulk-boundary correspondence, emerged as the most prominent representative of topological phases of matter. It turned out that they can be grouped into ten Altland-Zirnbauer symmetry classes (AZSCs), depending on their non-spatial symmetries (time-reversal, particle-hole, and sublattice or chiral) [14–16]. AZSCs also encompass thermal topological insulators or superconductors [17–19]. Subsequent inclusion of crystal symmetries in the classification scheme immensely diversified the landscape of topological phases that ultimately gave birth to topological quantum chemistry, nowadays routinely employed to identify topological crystals in nature [20–28]. Noticeably, model Hamiltonian for crystalline topological phases also belong to AZSCs, but often possess distinct topological invariants. In this realm, the following quest fuels our current venture. *Can a hybridization* (defined in the next paragraph) *between two topological insulators from different Altland-Zirnbauer symmetry classes foster* (possibly new) *topology?*

We offer an affirmative answer to this question by introducing the notion of hybrid symmetry class topological insulators (HSCTIs). Notice that all TIs (electrical or thermal) from AZSCs can be modeled by Dirac Hamiltonian with a momentum-dependent Wilson-Dirac mass [15, 16], manifesting band inversion around a time reversal invariant momentum (TRIM) point in the Brillouin zone (BZ). A hybridization between two TIs from distinct AZSCs occurs when they occupy orthogonal Cartesian hyperplanes and their band-inverted massive Dirac Hamiltonian mutually anticommute. As the resulting insulator stems from fusion or hybridization of two TIs from different AZSCs, we name it HSCTI. We show that the resulting HSCTI, also belonging to one of the AZSCs that is, however, distinct from the ones for the parent TIs, can nurture emergent topology. It should be noted that in time-reversal symmetry breaking systems the TRIM points in the BZ do not possess any special importance. But, in their lattice regularized models the band inversion still often occurs around the TRIM points, which appear at the high symmetry points of the BZ.

The construction of HSCTIs is showcased here from its simplest possible incarnation in three dimensions, stemming from the hybridization between a two-dimensional (2D) class AII $xy$ planar quantum spin Hall insulator (QSHI) and a one-dimensional (1D) class BDI $z$-directional Su-Schrieffer-Heeger insulator (SSHI). The top and the bottom surfaces of the resulting three-dimensional (3D) HSCTI, belonging to class A, then support 1D edge states of opposite chiralities, producing integer quantized Hall conductivity (in units of $e^2/h$) of opposite signs on these two surfaces. Thus, a 3D HSCTI differs from the presently known strong $Z_2$ and higher-order TIs, respectively supporting gapless surface states on six faces of a cube (first-order) [29] and $z$-directional hinge modes along with $xy$ surface states (second-order) [30] or eight corner modes (third-order) [31, 32]. See Fig. 1. It is also distinct from 3D axion insulators [33–36], displaying either half-quantized or non-quantized surface Hall conductivity. As such surface Hall conductivity due to a non-trivial axion angle $\theta_{\text{ax}}$, given by $\sigma_{\text{surface}}^{\text{axion}} = \theta_{\text{ax}}/(2\pi)$ (modulo $2\pi$) in units of $e^2/h$, can only be either half-quantized (for

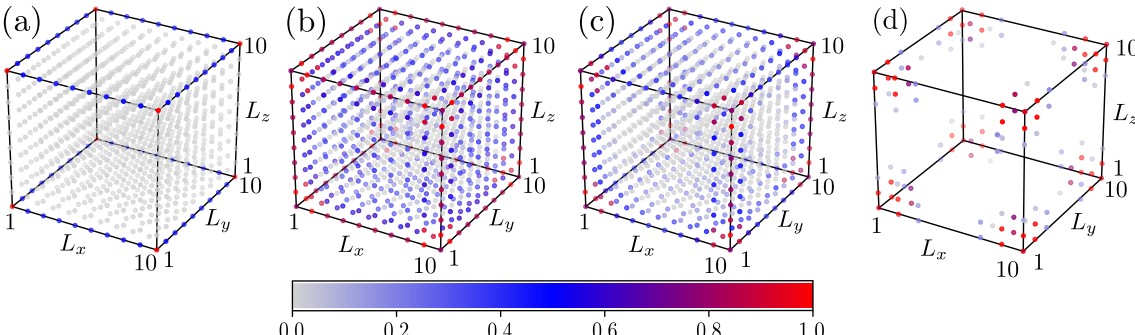

Figure 1: Normalized local density (LDOS) of states for (a) surface localized chiral edge states of HSCTI, (b) surface states of a first-order TI [29], (c) $z$-directional hinge and $xy$ surface state of a second-order TI [30], and (d) eight corner modes of a third-order TI [31, 32]. The lattice models for (b), (c) and (d) are shown in Appendix A. Throughout, LDOS on a site at position $r_i$ is defined as $\rho(r_i) = \sum_{j,\alpha} |\Psi_{i,\alpha}^{0,j}|^2$, where $i$ is the site index, summation over $j$ is restricted within the near zero energy manifold $\{|\Psi^{0,j}\rangle\}$ (indicated by the superscript '0') of the corresponding lattice-regularized Hamiltonian and $\alpha$ indicates the spin, orbital, and Nambu (for superconductors) degrees of freedom, for example.

$\theta_{\text{ax}} = \pi$) or non-quantized (for arbitrary $\theta_{\text{ax}} \neq \pi$) or trivial (for $\theta_{\text{ax}} = 0$), but not integer quantized as is the case for HSCTI. Therefore, the quantized surface Hall conductivity of HSCTI cannot be attributed to an axion angle. The HSCTI, yielding surface Chern insulators of opposite Chern numbers on the top and bottom surfaces only, should also be contrasted with recently proposed 'Embedded topological insulators', in which layers of Chern insulators get embedded between the slabs of trivial insulators in the bulk of a 3D system [37].

## 1.1 Organization

The rest of the manuscript is organized as follows. In Sec. 2, we introduce the model for 3D HSCTI, discuss its symmetries and phase diagram. Sec. 3 is devoted to the bulk-boundary correspondence, surface resolved Hall conductivity, topological invariant, and responses to lattice defects (dislocation) of 3D HSCTI. The notion of hybrid symmetry class topology is extended to encompass its crystalline and superconducting counterparts in Sec. 4. Concluding remarks, future directions and (meta)material perspectives of our proposals are presented in Sec. 5. Additional results and technical details are relegated to five Appendices. Specifically, in Appendix A we show the lattice-regularized models for 3D TIs, already known in the literature. Appendix B is devoted to the analytical derivation of the surface Hamiltonian for a 3D HSCTI and the computation of its topological invariant. Key details for the transport calculation are discussed in Appendix C. Addition details of the crystalline HSCTI are displayed in Appendix D. Construction of a four-dimensional HSCTI is shown in Appendix E.

## 2   3D HSCTI: Construction, symmetry and phase diagram

To arrive at the model Hamiltonian for the 3D HSCTI that captures all its salient features mentioned in the Introduction, consider first the Bloch Hamiltonian [5]

$$h_{\text{QSHI}}^{xy} = d_1(k)\Gamma_1 + d_2(k)\Gamma_2 + d_3(k)\Gamma_3. \tag{1}$$

The components of the $d$-vector for now are chosen to be $d_1(k) = t\sin(k_x a)$, $d_2(k) = t\sin(k_y a)$ and $d_3(k) = m_0 + t_0[\cos(k_x a) + \cos(k_y a)]$. Here $a$ is the lattice spacing. The hopping parameter $t$ is set to be unity. Mutually anti-commuting Hermitian $\Gamma$ matrices are $\Gamma_j = \sigma_3\tau_j$ for $j = 1, 2, 3$. The Pauli matrices $\{\tau_\mu\}$ ($\{\sigma_\mu\}$) operate on the orbital (spin) degrees of freedom with $\mu = 0, \cdots, 3$, with $\tau_0$ and $\sigma_0$ as identity matrices. Then the above model describes a QSHI in the $xy$ plane within the parameter regime $-2 < m_0/t_0 < 2$, featuring counter-propagating helical edge modes for opposite spin projections (class AII). When topological $h_{\text{QSHI}}^{xy}$ is implemented on a 3D cubic lattice without any tunneling in the $z$ direction, it supports a column of decoupled counter-propagating helical edge modes occupying the $xz$ and $yz$ planes. See Fig. 2(ai).

Next consider a second Bloch Hamiltonian [38–40]

$$h_{\text{SSHI}}^z = d_4(k)\Gamma_4 + d_5(k)\Gamma_5, \tag{2}$$

where $d_4(k) = t_1\sin(k_z a)$ and $d_5(k) = m_z + t_z\cos(k_z a)$. We set $t_1 = 1$. If $\Gamma_4$ and $\Gamma_5$ are anticommuting Pauli matrices, $h_{\text{SSHI}}^z$ describes a $z$-directional SSHI (class BDI). Within the parameter range $|m_z/t_z| < 1$, it supports topological zero energy modes, localized at its two ends. If we place such $z$-directional topological SSHIs on the sites of a square lattice on the $xy$ plane without any coupling between them, the resulting system features a collection of end point zero energy modes that occupies the entire top and bottom $xy$ surfaces. See Fig. 2(aii).

With the ingredients in hand, we now announce the Bloch Hamiltonian for a 3D HSCTI

$$h_{\text{HSCTI}}^{\text{3D}} = h_{\text{QSHI}}^{xy} + h_{\text{SSHI}}^z, \tag{3}$$

where now $\Gamma_4 = \sigma_1\tau_0$ and $\Gamma_5 = \sigma_2\tau_0$, that together with $\Gamma_1$, $\Gamma_2$ and $\Gamma_3$ constitute a set of five four-component Hermitian matrices, satisfying the Clifford algebra $\{\Gamma_j, \Gamma_k\} = 2\delta_{jk}$. Here $\delta_{jk}$ is the Kronecker delta function. Thus, $h_{\text{QSHI}}^{xy}$ and $h_{\text{SSHI}}^z$ anticommute with each other (hybridization). The $z$-directional SSHI acts as a mass for the edge modes of the $xy$ planar QSHI and vice versa. Then one component of $h_{\text{HSCTI}}^{\text{3D}}$ gaps out the topological modes of the other, except where both of them support topological gapless modes, namely along the edges on the top and bottom surfaces. See Fig. 2(aiii). But, the bulk is an insulator. Therefore, we realize a 3D TI by hybridizing two TIs, living on orthogonal Cartesian hyperplanes and belonging to different AZSCs, that manifests a bulk-boundary correspondence: a HSCTI, which also belongs to one of the ten AZSCs, namely class A in this case.

The model Hamiltonian for a 3D HSCTI $h_{\text{HSCTI}}^{\text{3D}}$ breaks (1) the time-reversal ($\mathscr{T}$) symmetry generated by $\mathscr{T} = \sigma_2\tau_1\mathscr{K}$, where $\mathscr{K}$ is the complex conjugation, and (2) the parity ($\mathscr{P}$) symmetry, generated by $\Gamma_3$ with $\mathscr{P} : k \to -k$. But, it preserves the composite $\mathscr{P}\mathscr{T}$ symmetry that guarantees a two-fold degeneracy of the conduction and valence bands of $h_{\text{HSCTI}}^{\text{3D}}$, respectively determined by the eigenspectra $\pm E(k)$, where $E(k) = [\alpha(k)]^{1/2}$ and $\alpha(k) = d_1^2(k) + \cdots + d_5^2(k)$.

Notice that $h_{\text{HSCTI}}^{\text{3D}}$, involving all *five* mutually anticommuting four-component Hermitian $\Gamma$ matrices, does not possess the sublattice or chiral symmetry, generated by a unitary operator that anticommutes with it. Rather it enjoys an anti-unitary particle-hole symmetry, generated by $\mathscr{A} = \sigma_0\tau_1\mathscr{K}$, such that $\{h_{\text{HSCTI}}^{\text{3D}}, \mathscr{A}\} = 0$ [41].

In the $(m_0/t_0, m_z/t_z)$ plane, a 3D HSCTI occupies a rectangular region bounded by $|m_0/t_0| < 2$ and $|m_z/t_z| < 1$, where both the parent insulators are topological. See Fig. 2(b). Furthermore, this topological regime fragments into two sectors for $-2 < m_0/t_0 < 0$ and $0 < m_0/t_0 < 2$, when the band inversion of the underlying QSHI takes place near the $\Gamma = (0,0)$ point ($\Gamma$ phase) and $M = (1,1)\pi/a$ point (M phase) of a 2D square lattice BZ, respectively [42]. Dislocation lattice defects are instrumental in distinguishing these two regimes about which more in a moment in the next section.

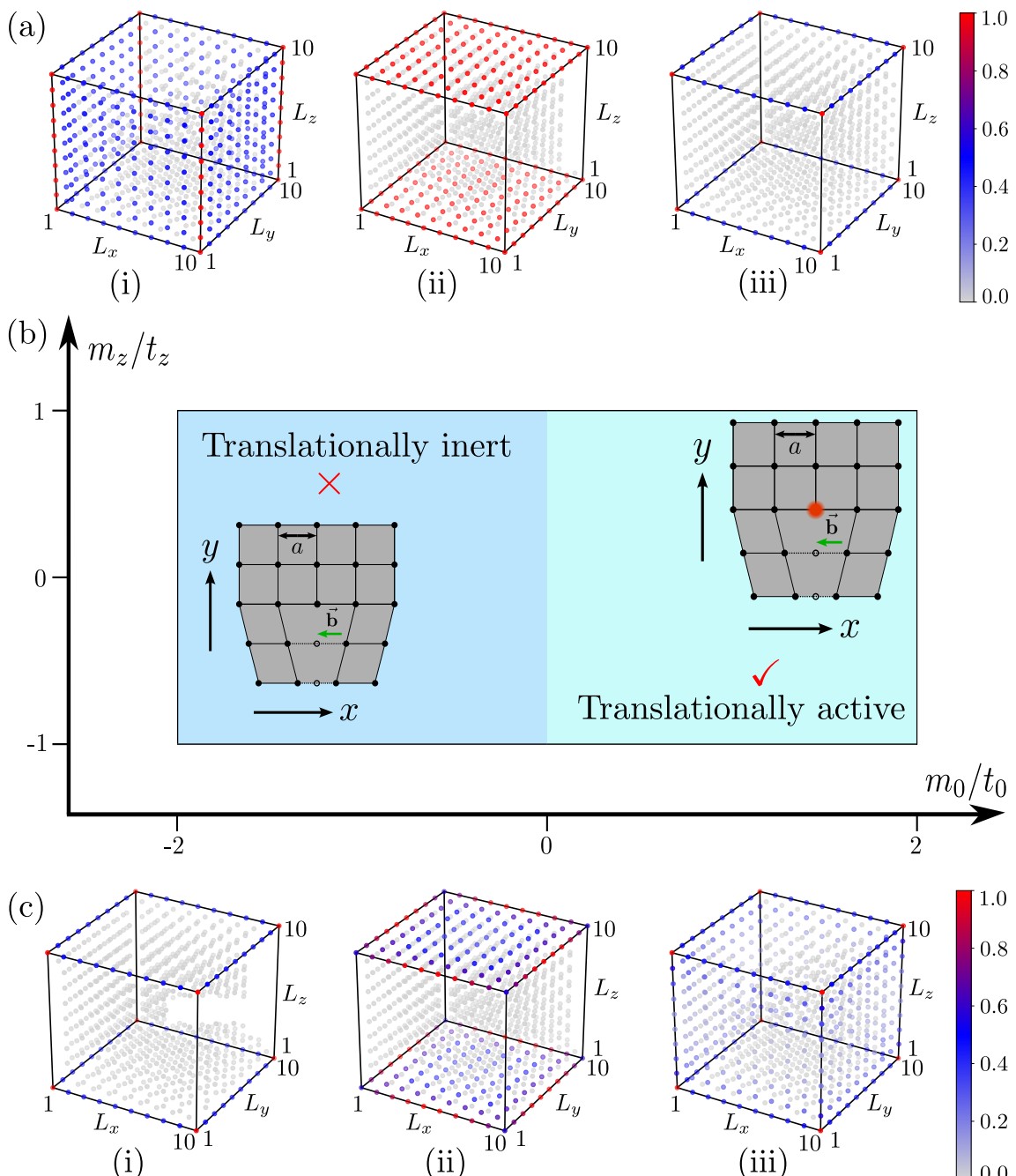

Figure 2: (a) Normalized local density of states for the (i) edge modes of decoupled QSHIs for $m_0/t_0 = 1$, (ii) endpoint modes for decoupled SSHIs for $m_z/t_z = 0$, and (iii) chiral edge modes of HSCTI for $m_0/t_0 = 1$ and $m_z/t_z = 0$ [same as Fig. 1(a)]. (b) Phase diagram of HSCTI in the $(m_0/t_0, m_z/t_z)$ plane. The translationally active (inert) phase supports (is devoid of) dislocation defect modes, shown schematically by the red circle. (c) Melting of (i) chiral edge modes [same as (aiii)] by tuning (ii) $m_0/t_0$ to **1.75** for a fixed $m_z/t_z = 0$ and (iii) $m_z/t_z$ to **0.75** for fixed $m_0/t_0 = 1.0$.

A HSCTI can be pushed out of the topological regime by tuning $m_0/t_0$ or $m_z/t_z$ or both. As only the ratio $m_0/t_0$ is tuned from the topological toward trivial regime, the edge modes living on the opposite sides of the top or bottom surfaces start to hybridize, as shown in Fig. 2(cii). By contrast, as we tune only $m_z/t_z$ out of the topological regime the edge modes residing on

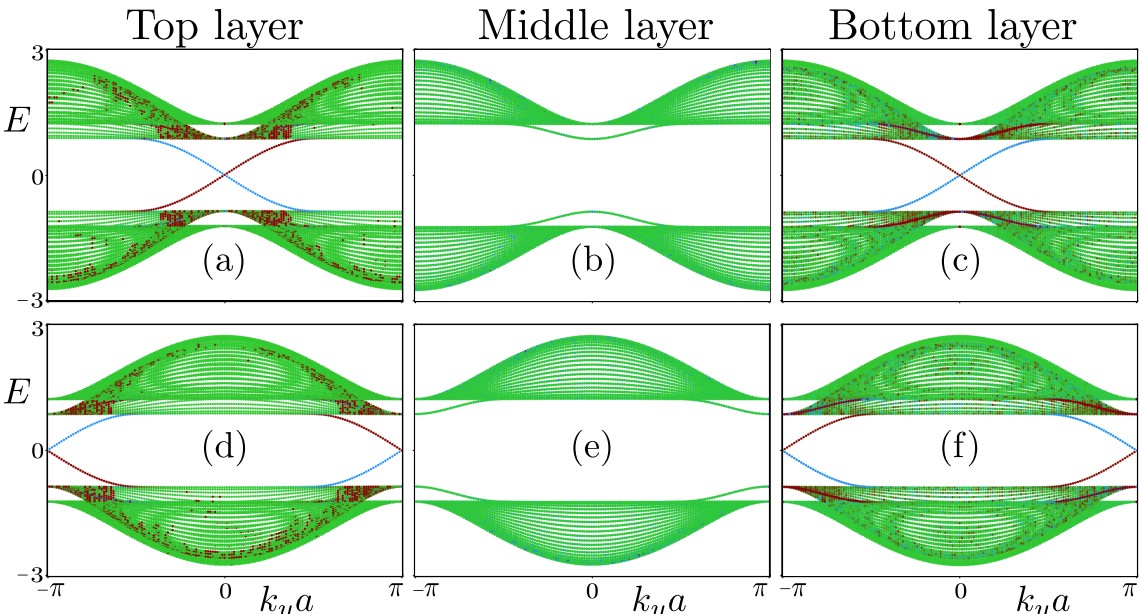

Figure 3: Layer resolved (in the $z$ direction) band structure of a 3D HSCTI with $k_y$ as a good or conserved quantum number and $L_x = L_z = 20$ for $m_z/t_z = 0$, and $m_0/t_0 = -1.0$ (upper panel) and $m_0/t_0 = 1.0$ (lower panel). Here blue (brown) and green indicate states localized near the left (right) edge and in the bulk of the system. Therefore, the top and bottom layers support counter-propagating chiral edge modes, while other layers are devoid of gapless states (such as the middle one). The localization of each mode (on the top and bottom surfaces, at the left and right edges, and in the bulk) is obtained from the spatial profile of the corresponding wavefunction, extracted by numerically diagonalizing the associated real space Hamiltonian.

the top and bottom surfaces mix through four side surfaces of the cube, as shown in Fig. 2(ciii). Once the system becomes a trivial insulator, there is no topological boundary modes.

## 3   3D HSCTI: Bulk-boundary correspondence, Hall effect, topological invariant, and lattice defects

The nature of the edge modes of the 3D HSCTI on the top and bottom surfaces can be anchored from the effective surface Hamiltonian. For simplicity, we consider a semi-infinite system with a hard-wall boundary at $z = 0$. When the region $z < 0$ ($z > 0$) is occupied by HSCTI (vacuum), the surface at $z = 0$ represents the top one. By contrast, when the region $z > 0$ ($z < 0$) is occupied by HSCTI (vacuum) the $z = 0$ surface corresponds to the bottom one. A straightforward calculation, shown in Appendix B, leads to the following surface Hamiltonian

$$h_{\text{surface}}^{\text{top/bottom}} = d_1(k)\beta_1 + d_2(k)\beta_2 \mp d_3(k)\beta_3, \tag{4}$$

where $k = (k_x, k_y)$. The newly introduced Pauli matrices $\{\beta_\mu\}$ operate on the space of two zero energy top/bottom surface states. With the chosen form of the $d$-vector, this Hamiltonian mimics the Qi-Wu-Zhang model for a square lattice quantum anomalous Hall insulator (QAHI) [43]. Therefore, the top and bottom surfaces of the 3D HSCTI harbor two-dimensional QAHIs with opposite first Chern numbers. On each surface the $\mathscr{T}$ symmetry is thus broken. In

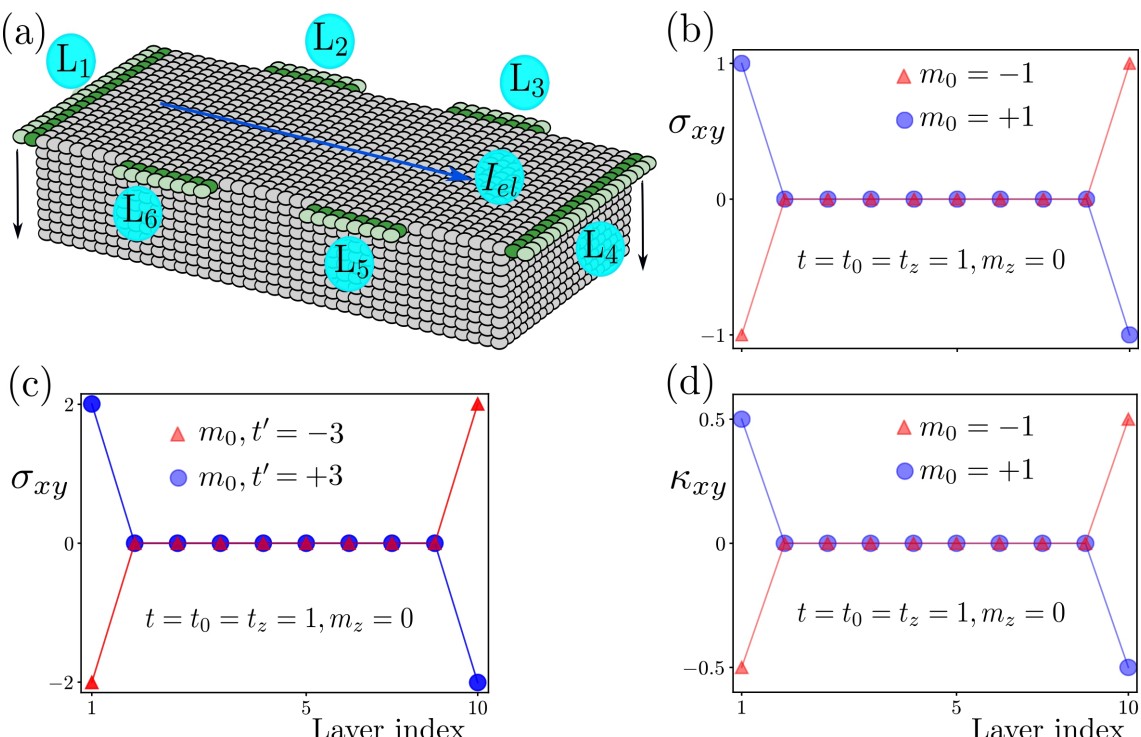

Figure 4: (a) Six-terminal setup for the layer-resolved Hall conductivity. (b) Layer-resolved electrical Hall conductivity ($\sigma_{xy}$) for HSCTI, showing its integer quantization (in units of $e^2/h$) on the top and bottom surfaces with opposite signs. (c) Same as (b), but for crystalline HSCTI with a parent QSHI featuring band inversion at the **X** and **Y** points of a 2D BZ. (d) Thermal Hall conductivity ($\kappa_{xy}$) for thermal HSCTI (superconductor), showing its half-integer quantization (in units of $\kappa_0 = \pi^2 k_B^2 T/(3h)$) at temperature $T = 0.01$ only the top and bottom surfaces with opposite signs. The parameter values chosen for numerical calculations are mentioned inside the corresponding subfigure.

addition, they also break the $\mathscr{P}$ symmetry, under which the top and bottom surfaces switch, as they foster QAHIs of opposite Chern numbers. Boundaries of a 3D HSCTI this way manifest the conserved composite $\mathscr{P}\mathscr{T}$ symmetry of its bulk. The surface Hamiltonian lacks the unitary chiral or sublattice symmetry ($S$), under which $\{h_{\text{surface}}^{\text{top/bottom}}, S\} = 0$. It possesses an emergent anti-unitary particle-hole symmetry ($C$), such that $\{h_{\text{surface}}^{\text{top/bottom}}, C\} = 0$, where $C = \beta_1 \mathscr{K}$ and $C^2 = +1$, since we neglected any particle-hole asymmetry, captured by $d_0(k)\Gamma_0$, where $d_0(-k) = d_0(k)$ and $\Gamma_0 = \sigma_0 \tau_0$ is the four-dimensional identity matrix, in the parent state. It plays no role in topology as long as the system remains an insulator and only shifts all the energy eigenvalues. Inclusion of such a term will give rise to $\beta_0 d_0(k)$ in the surface Hamiltonian, which is then devoid of the $C$ symmetry. Here $\beta_0$ is the two-dimensional identity matrix. Therefore, $h_{\text{surface}}^{\text{top/bottom}}$ and $h_{\text{HSCTI}}^{\text{3D}}$ belong to class A as the $\mathscr{T}$ symmetry is already broken [15, 16], a distinct AZSC from its parent QSHI (class AII) and SSHI (class BDI). Notice that the particle-hole asymmetric terms must be absent in a superconductor, which we discuss shortly, as the $C$ symmetry then corresponds to the microscopic charge-conjugation symmetry.

As the top and bottom surfaces host QAHIs of opposite Chern numbers, they feature counter-propagating chiral edge states. See Fig. 3. We consider a semi-infinite system with $k_y$ as good or conserved quantum number, and finite extensions in the $x$ and $z$ directions

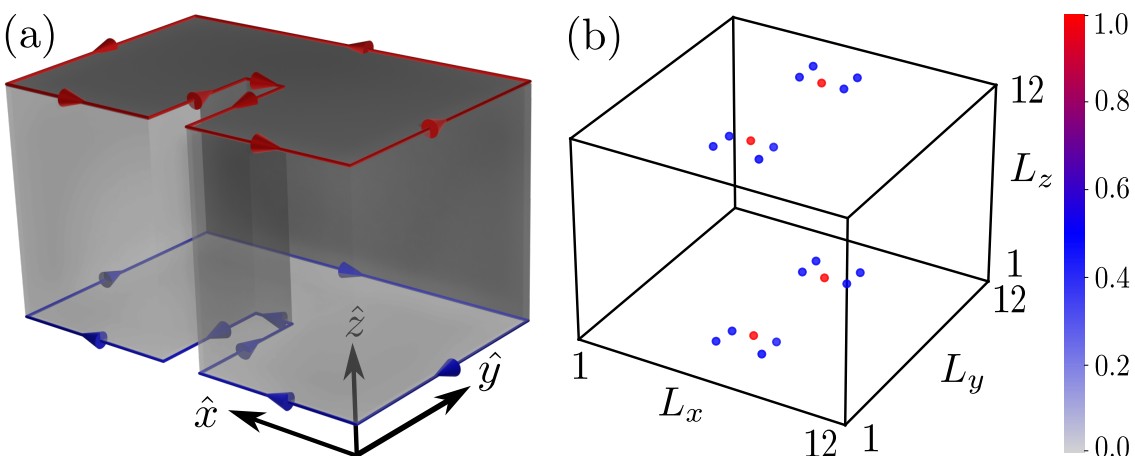

Figure 5: (a) A $z$-directional Volterra cut of a line of atoms in a 3D HSCTI crystal that creates new edges on $xy$ planes, supporting counter-propagating chiral edge modes only on the top and bottom surfaces. When these edges are pasted to create a line of edge dislocations, the edge modes hybridize and produce zero energy surface bound defect modes when the parent QSHI is in the **M** phase, for example. (b) Normalized local density of states for such defect modes with an edge dislocation-antidislocation pair and periodic boundary conditions in the $x$ and $y$ directions, for $m_0/t_0 = 1$ and $m_z/t_z = 0$. As 3D edge dislocation in (a) is constructed by stacking its 2D counterpart from Fig. 2(b), zero-energy defect modes on the top and bottom surfaces appear near the defect cores.

with open boundary conditions. For every $z$, we compute the band structure of a 3D HSCTI. Inside the topological regime of HSCTI, the top and bottom surfaces indeed feature counter-propagating edge modes crossing the zero energy at $k_y = 0$ ($\pm\pi/a$), when the underlying QSHI is in the **Γ** (**M**) phase. On the other hand, the middle layer is devoid of any chiral edge state. Surface localized chiral edge states also manifest through the layer-resolved integer quantized charge Hall conductivity ($\sigma_{xy}$), which we discuss next.

To compute the layer-resolved Hall conductivity in a 3D HSCTI, we consider a six-terminal Hall bar geometry. See Fig. 4(a). All the voltage and current leads are one-layer thick. An electrical current $I_{el}$ is passed between the leads $L_1$ and $L_4$. Then a transverse or Hall voltage develops between the leads $L_2$ and $L_6$, and $L_3$ and $L_5$. We numerically compute the Hall resistance $R_{xy}^{el} = (V_2 + V_3 - V_5 - V_6)/(2I_{el})$ using Kwant by attaching all the leads to a specific layer [44–46]. Here, $V_j$ is the voltage at the $j$th lead ($L_j$). The Hall conductivity is given by $\sigma_{xy} = (e^2/h)\left(R_{xy}^{el}\right)^{-1}$. The results are shown in Fig. 4(b). It shows that $\sigma_{xy}$ is quantized (in units of $e^2/h$) on the top and bottom surfaces, where they have opposite signs. It can be anchored by computing the first Chern number ($C$) of the surface Hamiltonian [Eq. (4)] as $\sigma_{xy} = Ce^2/h$. On any other layer $\sigma_{xy} = 0$. The overall sign of $\sigma_{xy}$ flips between the **Γ** and **M** phases of the parent QSHI. We also compute the two-terminal conductance ($G_{xx}$), but with the thickness of the leads equal to the sample thickness in the $z$ direction, yielding $G_{xx} = 2e^2/h$, confirming that there are exactly two topological edge modes in the $z$ direction. These quantized responses ($\sigma_{xy}$ and $G_{xx}$) are robust against weak and moderate disorder, while they vanish only in the strong disorder regime. Additional details of this computation are presented in Appendix C.

Although the parent 2D QSHI of class AII and 1D SSHI of class BDI, respectively possess non-trivial Pfaffian and Zak pahse, the resulting 3D HSCTI belongs to class A, which is devoid of any AZSC invariant in three dimensions [15,16]. Hence, their topological classification de-

mands a new invariant (beyond AZSC), which we develop now. It should nonetheless be noted that the 3D HSCTI features boundary modes along the edges of the top and bottom surfaces of cubic lattice, which are the boundaries of a boundary, where Altland-Zirnbauer topology is not operative. At the TRIM points $d_1(k) = d_2(k) = d_4(k) = 0$, and $h_{\text{HSCTI}}^{\text{3D}} = \Gamma_3 d_3(k) + \Gamma_5 d_5(k)$ can be brought into a block diagonal form after a suitable unitary rotation with $\Gamma_4 = \sigma_3\tau_1$ and $\Gamma_5 = \sigma_3\tau_2$. We then define a quantity $\hat{\varphi}_k = \varphi_k/|\varphi_k|$, where $\varphi_k = \tan^{-1}[d_5(k)/d_3(k)]$. By construction, $\hat{\varphi}_k = \pm 1$. The system then describes a TI only when

$$\Phi_z = \hat{\varphi}_{j,k_{z,1}^\star}\,\hat{\varphi}_{j,k_{z,2}^\star} = -1 \text{ and } \Phi_{xy} = \prod_j \hat{\varphi}_{j,k_{z,i}^\star} = -1, \tag{5}$$

for $i = 1$ and 2, where $k_z^\star = (0, \pi/a)$, $j = \Gamma, M, X, Y$ are the TRIM points of a 2D BZ with $X = (1,0)\pi/a$ and $Y = (0,1)\pi/a$. When $\Phi_z = -1$, the $z$-directional SSHI features band inversion along $k_z$ at all the TRIM points on the 2D BZ. On the other hand, when $\Phi_{xy} = -1$ the planar QSHI features band inversion at odd number of TRIM points on the 2D BZ for $k_z = 0$ and $\pi/a$. The TRIM band inversion point of the 2D BZ can be identified from the Pfaffian invariant [6]. On the other hand, if $\Phi_z$ or $\Phi_{xy}$ becomes $+1$, the system is a trivial insulator or crystalline HSCTI, which we discuss in the next section. Although HSCTI involves inversion of two Wilson-Dirac masses, it does not possess any multipole moment, such as the quadrupole and octupole moments [47–49]. Hence, a 3D HSCTI shall not be considered as a higher-order TI, even though it features topologically robust gapless modes on boundaries of codimension $d_c = d - d_B = 3 - 1 = 2 > 1$, defined as the difference between the dimensionality of the system ($d = 3$) and that for the gapless boundary modes ($d_B = 1$). On the same token, axion insulators [33–36], also supporting edge modes on the surfaces of a cubic lattice, does not correspond to a higher-order TI.

When the band inversion of the underlying QSHI occurs at a finite TRIM point ($K_{\text{inv}}^{\text{QSHI}}$), the 3D HSCTI becomes translationally active. Dislocation lattice defects, created by breaking the local translational symmetry in the bulk of a crystal, are instrumental to identify them in terms of topological modes bound to their cores. A screw dislocation fosters gapless modes only when the associated Burgers vector (**b**) pierces gapless surfaces [50–52]. As all the surfaces of a 3D HSCTI are gapped, screw dislocations do not host any metallic defect modes. A line of edge dislocation in three dimensions is characterized by **b** and the stacking direction ($\hat{\mathbf{s}}$). Only when $\hat{\mathbf{s}} = \hat{z}$ and $\mathbf{b} = a\hat{x}$ or $a\hat{y}$, the Burgers vector points toward gapless chiral edge states on the top and bottom surfaces. Once a line of atoms is removed, counter-propagating chiral edge states appear at the newly created edges on these two surfaces. See Fig. 5(a). Upon reconnecting these edges a 3D edge dislocation is created through the Volterra cut-and-paste procedure, and the edge modes hybridize. When $K_{\text{inv}}^{\text{QSHI}} \cdot \mathbf{b} = \pi$ (modulo $2\pi$) [42, 50–57], as is the case when the QSHI resides in the **M** or **XY** (introduced in the next section) phase, the nontrivial $\pi$ hopping phase around the defect core binds surface localized zero energy modes on the top and bottom surfaces only. See Fig. 5(b). This outcome can also be appreciated in the following way. When we 'paste' the edges, created during the Volterra 'cut' process, the hybridization or level repulsion between the counter-propagating 1D modes living on these two edges is captured by a domain-wall mass, whose sign changes across the line of missing atoms, and a uniform mass with no nontrivial spatial modulation, when $K_{\text{inv}}^{\text{QSHI}} \cdot \mathbf{b} = \pi$ (translationally active) and $0$ (translationally inert), respectively. Then according to the Jackiw-Rebbi mechanism [58], topological zero energy modes get pinned at the dislocation core, but only in the translationally active phase.

## 4  Crystalline HSCTI and superconductor

We now proceed to extend the construction of HSCTIs to encompass their counterparts that are protected by crystalline symmetries and superconductors. First, we focus on the former systems. With the following choices for the components of the $d$-vector

$$d_1(k) = S_x + C_x S_y, \quad d_2(k) = S_y + S_x C_y, \quad \text{and} \quad d_3(k) = m_0 - 2t' + t_0(C_x + C_y) + 2t'(C_x C_y),$$
(6)

where $S_j = \sin(k_j a)$ and $C_j = \cos(k_j a)$, the band inversion occurs at the **X** and **Y** points of the 2D BZ for $|m_0/t_0| > 2$ and $t'/t_0 > m_0/(4t_0)$, yielding a crystalline QSHI protected by the four-fold rotational ($C_4$) symmetry [21]. Such a phase for the QSHI is also named the XY phase, sometime also referred as the valley phase. The resulting HSCTI is then also protected by the $C_4$ symmetry. The layer-resolved Hall conductivity $\sigma_{xy} = \pm 2e^2/h$ on the top and bottom surfaces, respectively, as they host two counter-propagating chiral edge states. See Fig. 4(c). In this phase $\Phi_{xy} = +1$ as the band inversion for the QSHI occurs at an even number of TRIM points in the 2D BZ. But, the parity eigenvalues of the $xy$ planar QSHI model in this phase at the **X** and **Y** points are also $-1$, protected by the $C_4$ symmetry, while those at the $\Gamma$ and **M** points are $+1$. Although, the Pfaffian in this case is $+1$, the resulting XY QSHI phase is protected by the crystalline $C_4$ symmetry, and its invariant is $Z_2$ [21]. Additional details of the model are presented in Appendix D. Similar to the situation in the XY QSHI phase, $\hat{\varphi}_{j,k_{z,i}^\star} = -1$ for $j =$ **X** and **Y**, but $+1$ for $j = \Gamma$ and **M** with $i = 1, 2$ [see Eq. (5)] in the HSCTI with in-plane $C_4$ symmetry, yielding $\Phi_{xy} = +1$. This phase is distinct from a trivial insulator, where $\hat{\varphi}_{j,k_{z,i}^\star} = -1$ or $+1$ for $j = \Gamma, \mathbf{M}, \mathbf{X}, \mathbf{Y}$, as the band inversion therein occurs at the X and Y points simultaneously that are connected by the crystalline $C_4$ symmetry.

With a suitable $\Gamma$ matrix representation, $h_{\text{HSCTI}}^{\text{3D}}$ can describe a hybrid symmetry class topological superconductor. For example, when $\Gamma_1 = \eta_1 \sigma_0$, $\Gamma_2 = \eta_2 \sigma_3$ and $\Gamma_3 = \eta_3 \sigma_0$, where the set of Pauli matrices $\{\eta_\mu\}$ operates on the Nambu or particle-hole indices, $H_{\text{QSHI}}^{\text{xy}}$ describes a $p_x \pm i p_y$ paired state (class DIII), occupying the $xy$ plane and stacked in the $z$ direction. Now $t$ represents the pairing amplitude and $d_3(k)$ gives rise to a cylindrical Fermi surface when $|m_0/t_0| < 2$. In this basis $H_{\text{SSHI}}^z$ describes a $z$-directional Kitaev chain of Majorana Fermions (class D or BDI) that couples the layers of $p_x \pm i p_y$ superconductors. Physically, $d_4(k)$ ($d_5(k)$) describes a $p_z$-wave ($\mathcal{PT}$ symmetry breaking extended $s$-wave) pairing for $\Gamma_4 = \eta_2 \sigma_1$ and $\Gamma_5 = \eta_2 \sigma_2$. The resulting 3D thermal HSCTI or superconductor belongs to class D, which is also devoid of any invariant according to the AZSC. Nonetheless, it possesses a nontrivial invariant, see Eq. (5), which goes beyond the AZSC. Once again we note that as this paired state features Majorana edge modes on the top and bottom surfaces of a cubic lattice, i.e. along the boundaries of a boundary, AZ topology does not apply there, although $h_{\text{HSCTI}}^{\text{3D}}$ belongs to one of the ten AZSCs, namely class D in this case.

On the top and bottom surfaces, such a topological paired state supports 2D thermal Hall insulators of opposite Chern numbers. Their edge modes are constituted by counter propagating chiral Majorana fermions on opposite surfaces, each of which yields a half-quantized thermal Hall conductivity ($\kappa_{xy}$) in units of $\kappa_0$ at small temperature ($T \to 0$), where $\kappa_0 = \pi^2 k_B^2 T/(3h)$. Layer resolved numerical computation of $\kappa_{xy}$ in the six-terminal Hall bar geometry confirms this outcome and shows that it is indeed of opposite signs on the top and bottom surfaces. See Fig. 4(d). The two-terminal thermal conductance $G_{xx}^{th} = \kappa_0$ when the leads are attached to the entire system in the $z$ direction, in accordance with the fact that there are exactly two topological Majorana edge modes in the $z$ direction. The (half-)quantized thermal responses ($\kappa_{xy}$ and $G_{xx}^{th}$) are also robust against weak and moderate disorder, while $\kappa_{xy}, G_{xx}^{th} \to 0$ in the strong disorder regime. Details of this computation is shown in Appendix C. The edge dislocations with $\mathbf{b} = a\hat{x}$ or $a\hat{y}$ and $\hat{s} = \hat{z}$, in such a paired state support surface localized endpoint Majorana modes near the defect cores.

# 5   Discussions and outlooks

We outline a general principle of realizing HSCTIs from two distinct parent TIs that occupy orthogonal Cartesian hyperplanes and belong to different AZSCs. Explicitly discussed HSCTI, obtained via a hybridization between $xy$ planar QSHIs (class AII) and $z$-directional vertical SSHIs (class BDI), possesses a bulk topological invariant and manifests bulk-boundary correspondence by harboring surface QAHIs (class A), leaving its fingerprint on chiral edge states and layer-resolved quantized Hall effect. Our proposal thereby offers a unique approach to vision TIs at the boundaries of even higher-dimensional HSCTIs. For example, following the same principle a 3D class AII TI can be found on the boundary of a four-dimensional HSCTI, built from 3D class CII and 1D class BDI TIs, as shown in Appendix E. This route is distinct from the "dimensional reduction" of constructing a $d$-dimensional TI from a fixed AZSC $d+1$-dimensional one [59]. We also extend the jurisdiction of this proposal to systems, where at least one of the constituting parent TIs is protected by crystalline symmetry and to encompass superconducting states, featuring half-quantized surface thermal Hall conductivity. Existence of a plethora of strong and crystalline topological phases of matter (insulators and superconductors) of various dimensions and symmetries in nature [1–28] should therefore open an unexplored territory of HSCTIs (electrical and thermal) that in principle can be realized in quantum crystals and engineered in classical metamaterials, as long as their following general principles of construction are satisfied. (a) Individual insulators (electrical or thermal) must be in the topological phase, (b) they must occupy orthogonal Cartesian hyperplanes, (c) their corresponding Hamiltonian must anti-commute and (d) they must describe same type of quasiparticles (either charged or neutral). Recently, our proposed protocol has been generalized to construct 'Boundary topological insulators and superconductors' from the *hybridization* among topological insulators or superconductors from various AZSCs [60].

As $d_5(k)$ breaks both $\mathscr{P}$ and $\mathscr{T}$ symmetries, layered magnetic materials with columnar antiferromagnetic order in the stacking direction constitute an ideal platform to harness the candidate 3D HSCTI. With the recent discovery of (anti)ferromagnetic TI $MnBi_2Te_4$ [61–64] (possibly axionic), we are optimistic that HSCTI can be found in some available or newly synthesized quantum materials using the existing vast dictionary and catalog of magnetic materials [65], guided by topological quantum chemistry [24–27]. When such materials (once found) are doped, they can harbor thermal HSCTI or superconductors from local or on-site pairings, as by now it is well established that the local paired states often (if not always) inherit topology from parent normal state electronic bands, even when it is a trivial insulator in the presence of a Fermi surface, but in terms of neutral Majorana fermions [66–70]. We leave this topic for a future investigation.

As the model Hamiltonian for HSCTI is described in terms of only nearest-neighbor hopping amplitudes, it can be emulated in classical metamaterials, among which topolectric circuits [71–73] and mechanical lattices [74–76] are the two most prominent ones. In both setups existence of chiral edge modes of 2D TIs has been experimentally demonstrated from the unidirectional propagation of a weak (low energy) disturbance only along their edges [72,74–76]. Finally, we note that topological defect modes have been experimentally observed in quantum crystals [77–79] and mechanical lattices [80]. Thus, our predicted counter-propagating chiral edge modes on the opposite surfaces of the 3D HSCTI and surface localized dislocation bound states should be within the reach of currently available experimental facilities.

# Acknowledgments

S.K.D. was supported by the Startup Grant of B.R. from Lehigh University, and B.R. was supported by NSF CAREER Grant No. DMR-2238679. We thank Daniel J. Salib for useful discussions.

† Corresponding author: bitan.roy@lehigh.edu

# A Lattice models for 3D topological insulators

In Fig. 1, we compared the boundary modes of a 3D hybrid-symmetry class topological insulator (HSCTI) with the ones for 3D first-order, second-order, and third-order topological insulators (TIs). In this appendix, we write down the corresponding Bloch Hamiltonian. The Bloch Hamiltonian for a 3D first-order TI is given by [29]

$$
\begin{aligned}
h^{\text{3D}}_{\text{FOTI}} = t \left[ \sin(k_x a)\Gamma_1 + \sin(k_y a)\Gamma_2 + \sin(k_z a)\Gamma_3 \right] \\
+ \left\{ m_0 - 6t_0 + 2t_0 \left[ \cos(k_x a) + \cos(k_y a) + \cos(k_z a) \right] \right\} \Gamma_4.
\end{aligned}
\tag{A.1}
$$

The mutually anticommuting Hermitian $\Gamma$ matrices, satisfying the Clifford algebra $\{\Gamma_j, \Gamma_k\} = 2\delta_{jk}$, are given by

$$
\Gamma_1 = \tau_1\sigma_1, \ \Gamma_2 = \tau_1\sigma_2, \ \Gamma_3 = \tau_1\sigma_3, \ \Gamma_4 = \tau_3\sigma_0, \ \Gamma_5 = \tau_2\sigma_0.
\tag{A.2}
$$

The Pauli matrices $\{\tau_\mu\}$ and $\{\sigma_\mu\}$, respectively operate on the parity and spin indices, where $\mu = 0, \cdots, 3$. The above model is in the topological regime for $0 < m_0/t_0 < 12$ [51], when it features gapless topological surface states on all six surfaces of a cube. Their normalized local density of states for $m_0/t_0 = 2$ and $t = 1$ is shown in Fig. 1(b) of the main manuscript.

The Bloch Hamiltonian for a 3D second-order TI is given by [30]

$$
h^{\text{3D}}_{\text{SOTI}} = h^{\text{3D}}_{\text{FOTI}} + \Delta_1 \left[ \cos(k_x a) - \cos(k_y a) \right] \Gamma_5.
\tag{A.3}
$$

When $h^{\text{3D}}_{\text{FOTI}}$ is in the topological regime, a second-order TI is realized for arbitrarily weak or strong $\Delta_1$. When $m_0/t_0 = 2$, the band inversion of the 3D first-order TI occurs at the $\Gamma = (0,0,0)$ point of the 3D cubic BZ. Nontrivial $\Delta_1$, then gaps out the surface states of $h^{\text{3D}}_{\text{FOTI}}$ on the $xz$ and $yz$ surfaces, leaving four $z$-directional hinges gapless. In addition, it also leaves the top and bottom $xy$ surfaces gapless, as $\cos(k_x a) - \cos(k_y a)$ vanishes at the $\Gamma = (0,0)$ point of the top and bottom surface BZ. The normalized local density of these modes is shown in Fig. 1(c) for $m_0/t_0 = 2$, $t = 1$ and $\Delta_1 = 0.5$.

The Bloch Hamiltonian for a 3D third-order TI is given by [31, 32]

$$
h^{\text{3D}}_{\text{TOTI}} = h^{\text{3D}}_{\text{SOTI}} + \Delta_2 \left[ 2\cos(k_z a) - \cos(k_x a) - \cos(k_y a) \right] \Gamma_6.
\tag{A.4}
$$

As $h^{\text{3D}}_{\text{TOTI}}$ involves six mutually anticommuting Hermitian $\Gamma$ matrices, satisfying the Clifford algebra $\{\Gamma_j, \Gamma_k\} = 2\delta_{jk}$, their minimal dimensionality must be eight. They are now chosen to be

$$
\begin{aligned}
\Gamma_1 = \alpha_1\tau_1\sigma_1, \ \Gamma_2 &= \alpha_1\tau_1\sigma_2, \ \Gamma_3 = \alpha_1\tau_1\sigma_3, \ \Gamma_4 = \alpha_1\tau_3\sigma_0, \\
\Gamma_5 = \alpha_1\tau_2\sigma_0, \ \Gamma_6 &= \alpha_2\tau_0\sigma_0, \ \Gamma_7 = \alpha_3\tau_0\sigma_0.
\end{aligned}
\tag{A.5}
$$

The newly introduced Pauli matrices $\{\alpha_\mu\}$ with $\mu = 0, 1, 2, 3$ operate on the sublattice degrees of freedom and $\alpha_0$ is the two-dimensional identity matrix. On a second-order TI, any nontrivial $\Delta_2$ produces a third-order TI, featuring eight corner modes. Their normalized local density of states is shown in Fig. 1(d) for $m_0/t_0 = 2$, $t = 1$, $\Delta_1 = 1.0$ and $\Delta_2 = 1.0$.

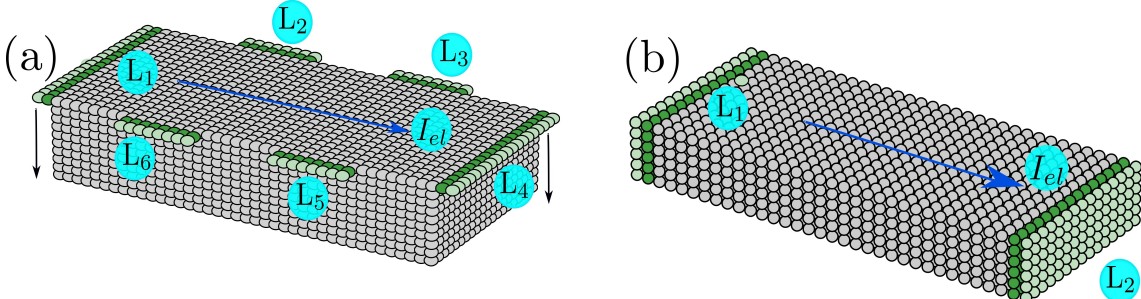

Figure 6: (a) Six-terminal Hall bar geometry [same as Fig. 4(a)], employed to compute the layer-resolved (electrical and thermal) Hall conductivity for a three dimensional system. The gray atoms correspond to the sites belonging to the three-dimensional scattering region, whereas the green ones indicate the sites that belong to the leads $L_1$, $L_2$, $L_3$, $L_4$, $L_5$, and $L_6$. A longitudinal electrical current $I_{el}$ is flowing across the leads $L_1$ and $L_4$ and correspondingly electrical Hall voltages are generated in the transverse leads. In case of the thermal Hall calculation, $I_{el}$ gets replaced with the longitudinal thermal current $I_{th}$, and thereby generating transverse thermal Hall voltage (temperature) in the vertical leads $L_2$, $L_3$, $L_5$, and $L_6$. (b) Two-terminal setup, employed to compute conductance (electrical and thermal) for a three-dimensional systems, where now the leads ($L_1$ and $L_2$) are connected to all the sites in the $z$ direction at the left and right edges of the scattering region (system). A longitudinal electrical (thermal) current $I_{el}$ ($I_{th}$) is flowing across the leads $L_1$ and $L_2$ while computing the two-terminal electrical (thermal) conductance.

## B   Surface states and surface Hamiltonian for 3D HSCTI

In this appendix, we show the analytical derivations of the surface Hamiltonian for the 3D HSCTI and compute the topological invariant of the surface TIs. The Bloch Hamiltonian for a 3D HSCTI, also shown in Eq. (3), reads as

$$h_{\text{HSCTI}}^{\text{3D}} = d_1(k)\Gamma_1 + d_2(k)\Gamma_2 + d_3(k)\Gamma_3 + d_4(k)\Gamma_4 + d_5(k)\Gamma_5. \tag{B.1}$$

The components of the five-dimensional $d$-vector are already announced and

$$\Gamma_1 = \sigma_3\tau_1, \ \Gamma_2 = \sigma_3\tau_2, \ \Gamma_3 = \sigma_3\tau_3, \ \Gamma_4 = \sigma_1\tau_0, \ \Gamma_4 = \sigma_2\tau_0. \tag{B.2}$$

To facilitate the computation of the surface states and the subsequent derivation of the surface Hamiltonian, we perform a unitary rotation by (without any loss of generality)

$$U = [\sigma_3\tau_1] \exp\left[-i\frac{\pi}{4}\sigma_2\tau_1\right] \exp\left[i\frac{\pi}{4}\sigma_2\tau_2\right] \exp\left[-i\frac{\pi}{4}\sigma_1\tau_2\right]. \tag{B.3}$$

Under this unitary rotation

$$U(\Gamma_1, \Gamma_2, \Gamma_3, \Gamma_4, \Gamma_5)U^\dagger = -(\sigma_1\tau_0, \sigma_2\tau_0, \sigma_3\tau_3, \sigma_3\tau_2, \sigma_3\tau_1), \tag{B.4}$$

such that the part of $h_{\text{HSCTI}}^{\text{3D}}$ that only depends on $k_z$ [namely, the terms appearing with $d_4(k) = t_1\sin(k_z a)$ and $d_5(k) = m_z + t_z\cos(k_z a)$] becomes block diagonal. The overall 'minus' sign is absorbed in the unimportant overall phase of the four-component spinor. Then

the $k_z$ dependent part of $h_{\mathrm{HSCTI}}^{\mathrm{3D}}$ takes the explicit form

$$h_{\mathrm{SSHI}}^{\mathrm{z}} = d_4(k)\sigma_3\tau_2 + d_5(k)\sigma_3\tau_1 = [d_4(k)\tau_2 + d_5(k)\tau_1] \oplus \{(-)[d_4(k)\tau_2 + d_5(k)\tau_1]\}$$
$$\equiv h_0^{\mathrm{up}}(k_z) \oplus h_0^{\mathrm{down}}(k_z).$$
(B.5)

For concreteness, we now focus on the lower two-dimensional block and ignore its overall 'minus' sign. Next we expand $h_0^{\mathrm{down}}(k_z)$ in powers of $k_z$ around $k_z = 0$, and take $k_z \to -i\partial_z$ as we seek to find the zero energy surface states $\Phi_0(z)$ on the top and bottom surfaces by breaking the translational symmetry in the $z$ direction. The pertinent second-order differential equation reads

$$h_0^{\mathrm{down}}(k_z \to -i\partial_z)\Phi_0(z) \equiv \left[ t_1(-i\partial_z)\tau_2 + \left( m_z + t_z + \frac{t_z}{2}\partial_z^2 \right)\tau_1 \right]\Phi_0(z) = 0, \quad \text{(B.6)}$$

where $\Phi_0^\top(z) = [\varphi_1(z), \varphi_2(z)]$ is a two-component spinor. As $\tau_3$ anticommutes with $h_0^{\mathrm{down}}(k_z)$, the surface zero energy states must be eigenstates of $\tau_3$, which allows for only two types of solutions: $\Phi_{0,1}^\top = [\varphi_1(z), 0]$ and $\Phi_{0,2}^\top = [0, \varphi_2(z)]$. The functional forms of $\varphi_1(z)$ and $\varphi_2(z)$ are obtained by solving the following differential equations respectively

$$t_1\partial_z\varphi_1(z) + \left[ m_z + t_z + \frac{t_z}{2}\partial_z^2 \right]\varphi_1(z) = 0 \text{ and } -t_1\partial_z\varphi_2(z) + \left[ m_z + t_z + \frac{t_z}{2}\partial_z^2 \right]\varphi_2(z) = 0.$$
(B.7)

Here we impose a hard wall boundary condition at $z = 0$, such that $\varphi_1(z = 0) = 0$ and $\varphi_2(z = 0) = 0$. After setting $t_1 = t_z = 1$ (without any loss of generality), the solutions read as

$$\varphi_1(z) = A\left( \exp[-z/\xi_+] - \exp[-z/\xi_-] \right), \text{ and } \varphi_2(z) = B\left( \exp[z/\xi_+] - \exp[z/\xi_-] \right), \text{ (B.8)}$$

where $A$ and $B$ are the normalization constants, and $\xi_\pm^{-1} = 1 \pm i\sqrt{1 + 2m_z}$ with $\Re(\xi_\pm^{-1}) > 0$.

The bottom surface of the HSCTI at $z = 0$ can be modeled as a hard wall boundary therein, such that the region $z > 0$ ($z < 0$) is occupied by HSCTI (vacuum). Then the wave function must vanish as $z \to \infty$, implying that $B = 0$ and the normalizable surface zero energy states is only $\varphi_1(z)$. An identical solution is obtained for $h_0^{\mathrm{up}}(k_z)$, yielding altogether two zero energy modes on the bottom surface, which in the four-component representation are given by

$$\left| \Psi_{0,1}(z) \right\rangle^\top = [\varphi_1(z), 0, 0, 0] \text{ and } \left| \Psi_{0,2}(z) \right\rangle^\top = [0, 0, \varphi_1(z), 0]. \quad \text{(B.9)}$$

The Hamiltonian on the bottom surface ($h_{\mathrm{surface}}^{\mathrm{bottom}}$) is now obtained by computing the matrix element of the remaining part of $h_{\mathrm{HSCTI}}^{\mathrm{3D}}$ which after the unitary rotation by $U$ [see Eq. (B.4)] takes the form

$$h_{\mathrm{QSHI}}^{\mathrm{xy}} = d_1(k)\sigma_1\tau_0 + d_2(k)\sigma_2\tau_0 + d_3(k)\sigma_3\tau_3 \quad \text{(B.10)}$$

in the two-dimensional space spanned by $\left| \Psi_{0,1}(z) \right\rangle$ and $\left| \Psi_{0,2}(z) \right\rangle$. Explicitly,

$$h_{\mathrm{surface}}^{\mathrm{bottom}} = \begin{pmatrix} \langle \Psi_{0,1}(z) | h_{\mathrm{QSHI}}^{\mathrm{xy}} | \Psi_{0,1}(z) \rangle & \langle \Psi_{0,1}(z) | h_{\mathrm{QSHI}}^{\mathrm{xy}} | \Psi_{0,2}(z) \rangle \\ \\ \langle \Psi_{0,2}(z) | h_{\mathrm{QSHI}}^{\mathrm{xy}} | \Psi_{0,1}(z) \rangle & \langle \Psi_{0,2}(z) | h_{\mathrm{QSHI}}^{\mathrm{xy}} | \Psi_{0,2}(z) \rangle \end{pmatrix}$$
$$= d_1(k)\beta_1 + d_2(k)\beta_2 + d_3(k)\beta_3.$$
(B.11)

The set of Pauli matrices $\{\beta_\mu\}$ operates on the subspace spanned by two zero energy surface modes. See Eq. (4).

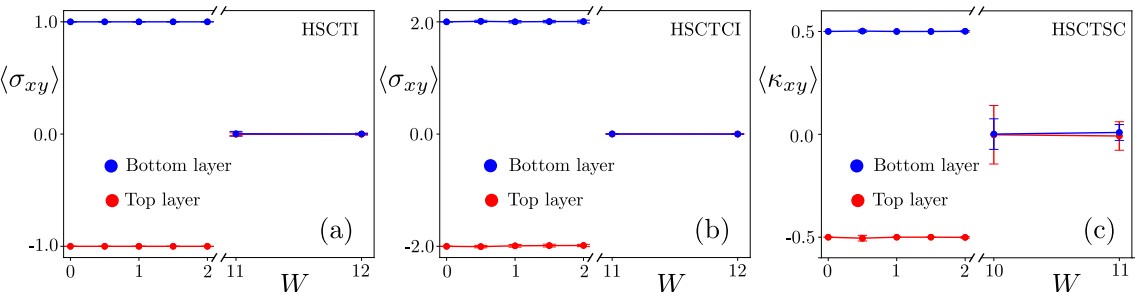

Figure 7: Disorder-averaged layer-resolved electrical Hall conductivity $\langle \sigma_{xy} \rangle$ for (a) hybrid symmetry class topological insulator (HSCTI) and (b) crystalline HSCTI, and (c) thermal Hall conductivity $\langle \kappa_{xy} \rangle$ for hybrid symmetry class topological superconductor (HSCTSC). Here $W$ denotes the strength of disorder, and $\langle \sigma_{xy} \rangle$ ($\langle \kappa_{xy} \rangle$) is measured in units of $e^2/h$ ($\kappa_0$). All the parameters are kept unchanged from the ones reported for panels (b), (c), and (d) of Fig. 4, respectively, and the system sizes for individual cases are reported in Appendix C. Disorder averaging is performed over 20 independent realizations in the weak disorder regime $W \leq 2$ and over 50 independent realizations in the strong disorder regime $W \geq 11$ for (a) and (b), and $W \geq 10$ for (c). Hence, these (half)-quantized responses are robust in the weak and moderate disorder regimes, while vanish only for strong disorder. Here the results are shown only for the top and bottom layers, whereas $\langle \sigma_{xy} \rangle = 0$ and $\langle \kappa_{xy} \rangle = 0$ in all other layers for any $W$, as in the clean limit. See Fig. 6(a) for the lead arrangement. Error bars correspond to the standard deviation.

The top surface at $z = 0$ is modeled by assuming that the region with $z < 0$ ($z > 0$) is occupied by HSCTI (vacuum). Then the zero energy surface states must vanish as $z \to -\infty$ besides at $z = 0$, implying $A = 0$ and now

$$\left| \Psi_{0,1}(z) \right\rangle^\top = [0, \varphi_2(z), 0, 0] \text{ and } \left| \Psi_{0,2}(z) \right\rangle^\top = [0, 0, 0, \varphi_2(z)]. \tag{B.12}$$

Identical calculation then leads to the following Hamiltonian on the top surface (see Eq. (4))

$$h_{\text{surface}}^{\text{top}} = d_1(k)\beta_1 + d_2(k)\beta_2 - d_3(k)\beta_3. \tag{B.13}$$

The first Chern number for the surface Hamiltonian is given by [12]

$$C = \int_{\text{BZ}} \frac{d^2 k}{4\pi} \left[ \partial_{k_x} \hat{d}(k) \times \partial_{k_y} \hat{d}(k) \right] \cdot \hat{d}(k), \tag{B.14}$$

where $\hat{d}(k) = d(k)/|d(k)|$. The integration is performed over the 2D surface BZ on the $xy$ plane. On the top and bottom surfaces $d(k) = (d_1, d_2, -d_3)(k)$ and $d(k) = (d_1, d_2, d_3)(k)$, respectively, confirming that the first Chern numbers on these two surfaces are of equal magnitude, but of opposite signs. When $d_1(k) = t \sin(k_x a)$, $d_2(k) = t \sin(k_y a)$ and $d_3(k) = m_0 + t_0[\cos(k_x a) + \cos(k_y a)]$, $C = -1$ and $+1$ on the bottom surface respectively for $0 < m_0/t_0 < 2$ and $-2 < m_0/t_0 < 0$.

## C  Layer-resolved Hall conductivity and two-terminal conductance

In this Appendix, we present the details of the transport geometry and calculations, reported in this work. All our numerical calculations were performed using Kwant transport package [44]. To begin with, we define a three dimensional (3D) finite system with the Hamiltonian shwon in Eq. (3) from the main manuscript for a system size of $L = 40$, $W = 20$, $H = 10$, where $L$, $W$, and $H$ represent the length in the $x$ direction, width in the $y$ direction, and height in the $z$ direction of the sample, respectively. For probing the layer dependent electrical Hall conductivity, we attach multiple leads on the corresponding layer that we are interested in. As shown in Fig. 6(a), for instance, we first attach all six semi-infinite leads to the top layer of the 3D system which are indicated by $L_1$, $L_2$, $L_3$, $L_4$, $L_5$, and $L_6$. A longitudinal current $I_{el}$ then flows between the leads $L_1$ and $L_4$. We place the other four leads ($L_2$, $L_3$, $L_5$ and $L_6$) in the transverse direction to probe the Hall voltages generated in the system. We repeat the same lead attachment procedure as we change the layer which are depicted by the black arrows in Fig. 6(a). Equipped with this scattering region and leads setup, we can then get the scattering matrix as

$$S = \begin{pmatrix} r & t' \\ t & r' \end{pmatrix}, \tag{C.1}$$

where $r$, $r'$ and $t$, $t'$ are the reflection and transmission blocks of the scattering matrix. Since we have six leads attached to the system, the scattering matrix in Eq. (C.1) has a $6 \times 6$ block structure, capturing all the possible matrix elements between different leads. In our calculation, leads have the same Hamiltonian as that of the scattering region. It is important to note that both in electrical and thermal Hall calculations, the leads are selectively attached to only one single layer, however, the scattering region consists of all layers, meaning that it spans the entire 3D sample.

Let us first focus on how one can calculate six terminal electrical Hall response. As there is a current flow across the system in presence of an applied electric field $E$, the current density ($j$)-electric field relation reads $j_a = \sum_b \sigma_{ab} E_b$, where $\sigma_{ab}$ is called the conductivity tensor. Since in our setup [Fig. 6(a)], the current is only flowing along the $x$ direction between the leads $L_1$ and $L_4$, one can probe the off-diagonal components of the conductivity tensor from the voltage drop between $L_2$, $L_3$, $L_5$, and $L_6$. Finally, the Hall conductivity can be written as

$$\sigma_{xy}^H = \frac{j_x E_y}{E_x^2 + E_y^2}, \tag{C.2}$$

where, $E_x = (V_2 - V_3)/L_{23}$, $E_y = (V_2 + V_3 - V_5 - V_6)/(2W)$ and $L_{23} = L/5$. Here $V_{i=1,\cdots,6}$ are the voltages developed in all the leads (numbered accordingly).

Regarding the computation of the layer-resolved Hall conductivity in a crystalline HSCTI a comment is due at this stage. It is important to emphasize that the crystalline topological insulating phase (XY phase) is protected by the $C_4$ rotational symmetry about the $z$ axis, due to which our 3D scattering region now maintains a $C_4$ symmetric square shaped planes for any value of $H$. In this case, for the calculation of the electrical Hall conductivity, we take the system size of $L = 80$, $W = 80$ and $H = 10$.

For the calculation of the thermal Hall conductivity ($\kappa_{xy}$), we consider the similar transport setup with the scattering region and leads, except the fact that now there is a longitudinal thermal current ($I_{th}$) flows between leads $L_1$ and $L_4$, and thereby generating thermal Hall voltages in the transverse leads [45]. In this case, we use the thermal current-temperature relation $\mathbf{I}_{th} = \mathbf{A}_{th}\mathbf{T}$, where $\mathbf{I}_{th}^\top = (I_{th}, 0, 0, -I_{th}, 0, 0)$ and $\mathbf{T}^\top = (-\Delta T/2, T_2, T_3, \Delta T/2, T_5, T_6)$. Here $T$ is the temperature of the scattering region, and the matrix elements of $A$ can be written as

$$A_{th,ij} = \int_0^\infty \frac{E^2}{T} \left( -\frac{\partial f(E,T)}{\partial E} \right) \left[ \delta_{ij}\mu_j - \mathrm{Tr}(t_{ij}^\dagger t_{ij}) \right] dE. \tag{C.3}$$

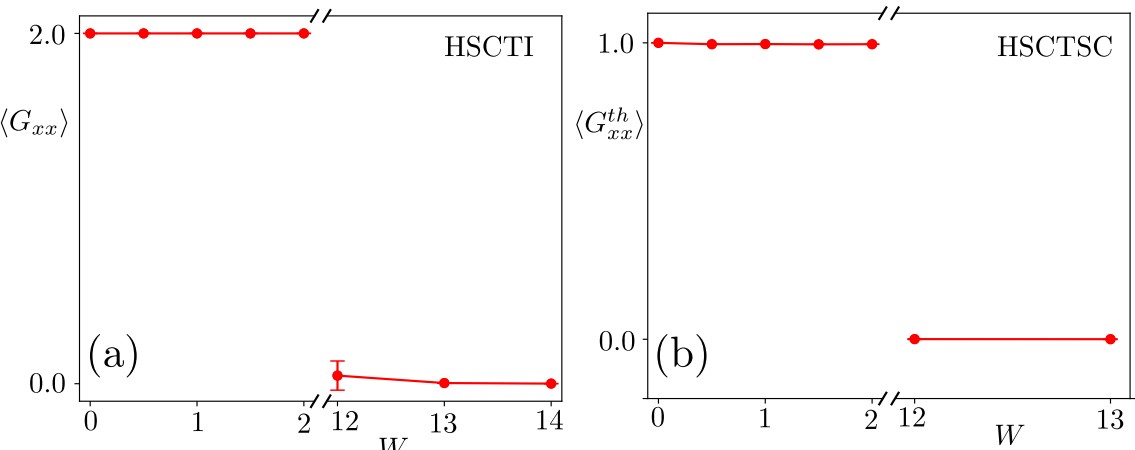

Figure 8: Disorder-averaged two-terminal (a) electrical conductance ($\langle G_{xx} \rangle$) for HSCTI and (b) thermal conductance ($\langle G_{xx}^{th} \rangle$) for hybrid symmetry class topological superconductor (HSCTSC). Here $W$ denotes the strength of disorder, and $\langle G_{xx} \rangle$ ($\langle G_{xx}^{th} \rangle$) is measured in units of $e^2/h$ ($\kappa_0$). All the parameters are kept unchanged from the ones reported for panels (b) and (d) of Fig. 4, respectively, and the system sizes for individual cases are reported in Appendix C. The disorder averaging is performed over 20 independent realizations in the weak disorder regime ($W \leq 2$) and over 50 independent realizations in the strong disorder regime ($W \geq 12$). Hence, these quantized responses are robust in the weak and moderate disorder regimes, while vanish for strong disorder. See Fig. 6(b) for the lead arrangement. Error bars correspond to the standard deviation.

In the above equation, $\mu_j$ denotes the number of conducting channels in the $j$th lead, $f(E, T) = [1 + \exp[E/(k_B T)]]^{-1}$ is the Fermi-Dirac distribution function, $\mathbf{t}_{ij}$ is the transmission part of the scattering matrix between the leads $L_i$ and $L_j$, and the trace ($\mathbf{Tr}$) is taken over the conducting channels. Next, we obtain the thermal Hall resistance as $R_{xy}^{th} = (T_2 + T_3 - T_5 - T_6)/(2I_{th})$, and inverting it yields $\kappa_{xy} = \pi^2 k_B^2 T/(3h) \left( R_{xy}^{th} \right)^{-1} \equiv \kappa_0 \left( R_{xy}^{th} \right)^{-1}$. Note that, we take the average over different terminals to avoid contact resistance effects. We set $k_B = h = 1$ and $T = 0.01$ for all our calculations. The system size for the thermal Hall conductivity computation is $L = 40$, $W = 20$, and $H = 10$.

Using the identical approach, we now compute layer-resolved electrical and thermal Hall conductivities in disordered systems. We only consider pointlike random charge impurities, the dominant source of elastic scattering in any real material. The presence of random charge impurities enter the Hamiltonian as $V(r)\sigma_0\tau_0$ for HSCTI and crystalline HSCTI, and as $V(r)\eta_3\sigma_0$ (in the Nambu basis) for hybrid symmetry class topological superconductor. In both cases $V(r)$ couples to local density, which is uniformly and randomly distributed within the range $[-W/2, W/2]$ on every site belonging to the three-dimensional scattering region, and $W$ is the disorder strength. The results are displayed in Fig. 7, showing that layer-resolved (half-)quantized Hall responses and thus the topology of the bulk system are robust against weak and moderate disorder, while disappearing in the strong disorder regime. See Fig. 7.

Next, we compute two-terminal electrical ($G_{xx}$) and thermal ($G_{xx}^{th}$) conductance for hybrid symmetry class topological insulators and superconductors, respectively. We now attach two thick leads to the left and right sides of all the layers of the three-dimensional scattering region in the $z$ direction [Fig. 6(b)]. An electrical (A thermal) current $I_{el}$ ($I_{th}$) is then passed across the scattering region from lead $L_1$ to lead $L_2$, while computing $G_{xx}$ ($G_{xx}^{th}$) be-

tween these two leads. In this scenario, $G_{xx}$ and $G_{xx}^{th}$ can only capture the number of modes propagated between the longitudinal leads, however through the entire 3D system or scattering region, thereby displaying the topological invariant or quantization. The system size is $L = 40$, $W = 40$ and $H = 6$ for the calculation of both $G_{xx}$ and $G_{xx}^{th}$. We have shown our results for a fixed $m_0 = 1$, and rest of the parameters are kept same as in the layer resolved electrical and thermal Hall computation.

The longitudinal two-terminal electrical conductance is given by $G_{xx} = \text{Tr}(t_{ij}^{\dagger} t_{ij})$. Here $i, j = 1, 2$ are the lead numbers and the trace ($\text{Tr}$) is taken over the transmission channels. The computation is performed at the zero temperature and within the energy window $|E| \leq 2$ with $201$ grid points. The disorder averaged two-terminal electrical conductance $\langle G_{xx} \rangle$ is obtained after averaging over $20$ ($50$) independent disorder realizations for small (large) disorder strength with $W \leq 2$ ($W \geq 12$). For HSCTI, we find $G_{xx} = 2e^2/h$ in the weak and moderate disorder regimes, confirming that there are exactly two topological edge modes in the entire system, localized on the top and bottom surfaces (revealed by layer-resolved electrical Hall conductivity) which are robust against weak and moderate disorder. By contrast, the system becomes a trivial insulator for strong disorder, where $G_{xx} = 0$. See Fig. 8(a).

In the same spirit, we compute the two-terminal thermal conductance for hybrid symmetry class topological superconductor as $G_{xx}^{th} = \text{Tr}(t_{ij}(T)^{\dagger} t_{ij}(T))$, where $T = 0.01$ is the temperature of the scattering region, and the computation is performed within the energy window $|E| \leq 0.5$ with $201$ grid points. In this situation as well we find that $G_{xx}^{th} = \kappa_0$ in the weak and moderate disorder regimes, confirming that there are exactly two topological Majorana edge modes in the entire system, localized on the top and bottom surfaces (revealed by the layer-resolved thermal Hall conductivity) which are robust against weak and moderate disorder. On the other hand, the system becomes a trivial thermal insulator in the strong disorder regime, where $G_{xx}^{th} = 0$. See Fig. 8(b).

For a similar calculation of $G_{xx}$ in a crystalline HSCTI, the scattering region must be at least $80 \times 80$ in the $xy$ plane (see the system size for the layer-resolved $\sigma_{xy}$), which is then needed to be attached to at least a few layer thick three-dimensional leads. The numerical analysis then becomes extremely unstable due to large memory requirement, which is unfortunately far beyond our numerical resources. For this reason, at present we cannot compute $G_{xx}$ even in a clean crystalline HSCTI, leaving aside its disorder averaging. Nevertheless, based on ample convincing evidences we present for all the other cases, we can safely conclude that $G_{xx} = 4e^2/h$ in the clean system and in the weak and moderate disorder regimes, while it will only vanishes for strong disorder.

## D Crystalline HSCTI: Details

In this appendix, we present details of the construction of HSCTI that is protected by crystalline symmetry. As the HSCTI is constructed from the hybridization between a $xy$-planar QSHI and a $z$-directional SSHI, we can find its crystalline version (a 3D crystalline HSCTI) when the underlying QSHI is in the crystalline phase. Such a phase of matter can be captured by modifying the associated $d$-vector to the following one [21]

$$d_1(k) = t[\sin(k_x a) + \cos(k_x a)\sin(k_y a)], \quad d_2(k) = t[\sin(k_y a) + \sin(k_x a)\cos(k_y a)],$$
$$\text{and } d_3(k) = m_0 - 2t' + t_0[\cos(k_x a) + \cos(k_y a)] + 2t'\cos(k_x a)\cos(k_y a). \tag{D.1}$$

But, $d_4(k)$ and $d_5(k)$ are left unchanged. The phase diagram of this model is shown in Fig. 9(a). With this choice of the $d$-vector, the QSHI model features band inversion simul-

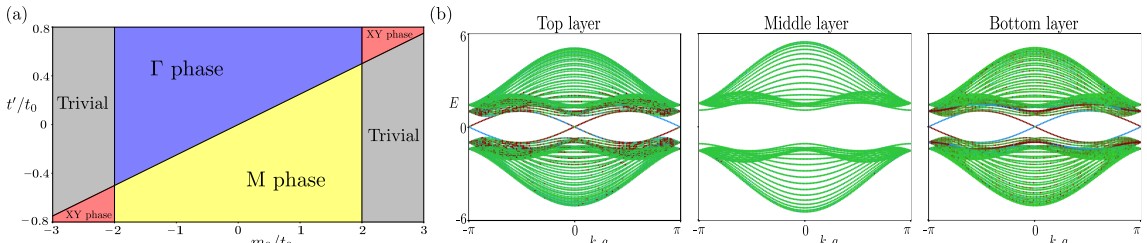

Figure 9: (a) Phase diagram of a 2D quantum spin Hall insulator model with the three components of the $d$-vector shown in Eq. (D.1). The band inversion in the $\mathbf{\Gamma}$ phase, $\mathbf{M}$ phase, and $\mathbf{XY}$ phase take place at the $\mathbf{\Gamma} = (0,0)$, $\mathbf{M} = (1,1)\pi/a$, and simultaneously at the $\mathbf{X} = (1,0)\pi/a$ and $\mathbf{Y} = (0,1)\pi/a$ points of the 2D square lattice BZ. (b) Layer-resolved (in the $\mathbf{z}$ direction) band structure of a 3D crystalline HSCTI on the top, middle, and bottom layers with $k_y$ as a good or conserved quantum number and $L_x = L_z = 20$ for $t = t_1 = 1$, $m_z/t_z = 0$, $m_0/t_0 = 3.0$, $t'/t_0 = 1.0$ [Eq. (D.1)]. Here blue (brown) and green indicate states localized near the left (right) edge and in the bulk of the system. It shows that two chiral edge modes live on the top and bottom surfaces only, while the other layers (such as the middle one) is devoid of any boundary modes. The chiral edge modes on the top and bottom surfaces are counter-propagating. The localization of each mode (on the top and bottom surfaces, at the left and right edges, and in the bulk) in (b) is obtained from the spatial profile of the corresponding wavefunction, extracted by numerically diagonalizing the associated real space Hamiltonian.

taneously at the $\mathbf{X}$ and $\mathbf{Y}$ points of the 2D BZ ($\mathbf{XY}$ phase), when

$$\frac{t'}{t_0} > \frac{1}{4}\frac{m_0}{t_0} \quad \text{and} \quad |m_0/t_0| > 2.$$

The surface Hamiltonian of this model are same as in Eq. (B.11) and Eq. (B.13), but in terms of the modified $d_1(k)$, $d_2(k)$ and $d_3(k)$, given in Eq. (D.1). The first Chern number of the surface Hamiltonian $C = -2$ $(+2)$ on the bottom surface when $m_0/t_0 > 2$ $(m_0/t_0 < -2)$ and $t'/t_0 > m_0/(4t_0)$. Once again the Chern number on the top surface is exactly opposite of that on the bottom surface. The layer-resolved (in the $\mathbf{z}$ direction) band structure with $k_y$ as good quantum number shows the existence of two chiral edge modes crossing the zero energy at $k_y = 0$ and $k_y = \pi/a$ [Fig. 9(b)] only on the top and bottom surfaces. On these two opposite surfaces the chiral edge modes are counter-propagating. By contrast, the other layers is devoid of any gapless topological modes (such as the middle one).

## E Four-dimensional (4D) HSCTI

Our construction of the 3D HSCTI can readily be generalized to four-dimensions, which we show in this appendix. It allows us to realize a 3D class AII TI on the 3D hypersurface of a 4D HSCTI constructed by hybridizing a 3D class CII TI, occupying the $xyz$ space and a 1D SSHI in the $w$ direction. Note that $x$, $y$, $z$, and $w$ are four mutually orthogonal principal

Cartesian axes in four dimensions. The corresponding Bloch Hamiltonian read as [15]

$$h_{\text{CII}}^{\text{xyz}} = t \left[ \sin(k_x a)\Gamma_1 + \sin(k_y a)\Gamma_2 + \sin(k_z a)\Gamma_3 \right]$$
$$+ \left\{ m_0 - 6t_0 + 2t_0 \left[ \cos(k_x a) + \cos(k_y a) + \cos(k_z a) \right] \right\} \Gamma_4, \quad (\text{E.1})$$
$$\text{and } h_{\text{SSHI}}^{\text{w}} = t_1 \sin(k_w a)\Gamma_5 + [m_w + t_w \cos(k_w a)]\Gamma_6.$$

The mutually anticommuting Hermitian $\boldsymbol{\Gamma}$ matrices can be chosen from the representation shown in Eq. (A.5). The Bloch Hamiltonian for the resulting 4D HSCTI then reads as $h_{\text{HSCTI}}^{\text{4D}} = h_{\text{CII}}^{\text{xyz}} + h_{\text{SSHI}}^{\text{w}}$.

On the top and bottom 3D hypersurfaces of such a 4D HSCTI in the $w$ direction, we recover class AII TIs, with the corresponding Hamiltonian analogous to the one shown in Eq. (A.1) in terms of four-component Hermitian $\boldsymbol{\Gamma}$ matrices [Eq. (A.2)]. This calculation identically follows the one shown in Appendix B.

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
