# Peer review of "Hybrid symmetry class topological insulators"

_SciPost Physics_

## Round 1 · Referee Report · Jasper van Wezel (Referee 1) · 2024-5-1

Strengths
- well-written
- topical
- broadly applicable
Weaknesses
- presentation / claims misleading in several points
- detailed example but no general proof
Report
The authors introduce the concept of "hybrid symmetry class topological insulators" and work out one particular example in detail.
The manuscript is very well-written, and the analysis of the example is both clear and thorough. Moreover, the idea of combining two types of topological insulators to create a third is certainly useful and seems versatile and broadly applicable in follow-up work. The content is therefore suitable for SciPost Physics.
However, I believe the way this idea is presented in the current manuscript is misleading in several central aspects. I would like to suggest that the authors phrase these differently.
In particular: 1) the authors claim several times that their results go beyond the Altland-Zirnbauer classification. However, this is simply not true, and moreover, not possible within the current setup. The Hamiltonian of Eq. (3) is a 3D Hamiltonian for non-interacting spin-1/2 particles in a two-orbital lattice. As written by the authors themselves below Eq. (4), this Hamiltonian falls in class A of the Altland-Zirnbauer classification, which in 3D is always trivial. This agrees with the observation of the authors that introducing a single (infinitely large) boundary into the periodic system along any direction results in an absence of surface states. The Altland-Zirnbauer classification does not say anything about the boundaries of boudaries (see also point 2 below), so as far as I can see the model introduced by the authors neatly fits into the Altland-Zirnbauer paradigm.
2) The authors also claim that the model of Eq. (3) is not a higher-order toplogical insulator (HOTI). However, introducing a boundary of the boudary (that is, cutting the periodic system in two orthogonal directions) may result in the emergence of edge states along the hinges. This clearly shows, as also argued by the authors, that the 3D system is a trivial insulator, while its 2D surface Hamiltonian (in one direction) is a topological insulator. A trivial insulator whose edges are topological insulators is the textbook definition of a second order topological insulator. I therefore do not understand why the authors insist their model is not a HOTI.
3) The authors present their results in very general terms, suggesting that all results are generic and that their construction will work for any combination of topological systems. However, they analyse only a single model in detail and discuss broader applications only in terms of extensions of that one model. There is no proof that any of the presented results are applicable more generally. In fact, some results certainly are not. For example, the topological invariant of eq. (5) should in general be Z-valued, rather than Z2-valued (being an invariant for a QAHI), and already fails to apply to the system with C4 symmetry in section 4 (as mentioned by the authors).
I would suggest that the authors are open about these aspects: their work provides a methodology for constructing HOTIs from lower-dimensional topological systems, and they analyse one particular example in great detail. This is a worthwhile result, without the need to claim anything more.
I also noticed some minor details:
- In the introduction, band inversions at TRIM are mentioned in a sentence referring to all AZ classes. Since TRIM do not have any special meaning in TRS-broken classes, this statement should be revised.
- The terminology "hybrid" can be confusing: the authors refer to hybridization between two terms in the Hamiltonian, rather than hybridization between spatially separated systems with an interface. It would be good to make this explicit early on.
- In the first sentence of section 2, the author mention "the" model, where they probably mean "a" model.
- It would be good to include details of the KWANT algorithm in appendix C.
Requested changes
- Rephrase the presentation of the results to avoid misleading claims.
- Consider minor points mentioned in the report.
Recommendation
Ask for major revision
Author: Bitan Roy on 2025-04-23 [id 5408]
(in reply to Report 1 by Jasper van Wezel on 2024-05-01)
We thank the referee for the report. The “Strengths” of the manuscript, “well-written”, “topical”, and “broadly applicable”, as identified by the referee, are encouraging. The referee concisely summarized our keys results, giving us confidence regarding the clarity of the presentation. We thank the referee for considering the manuscript to be “suitable for SciPost Physics”. Below we respond to the concerns of the referee, which also address the “Weaknesses” identified by the referee and make adequate changes to the manuscript.
1. We agree that any Hamiltonian in any dimension must belong to one of the AZSC. See a similar comment from Referee 2, our responses to it, and changes we make to the manuscript. We also agree that AZSC does not apply to TIs that support topological modes living on the boundary of a boundary, which we now clarify.
2. If we go by the geometry or co-dimension of the boundary modes of HSCTI, then it is tempting to conclude that 3D HSCTI corresponds to a second-order TI, featuring 1D edge modes with co-dimension 3-1=2. See also a comment from Referee 3, our responses to it and changes to the manuscript. But a HOTI must have a non-trivial higher-order moment, like quadrupolar or octupolar moment. But our HSCTI model does not possess any such moment. Thus, it is not HOTI. Notice that if we classify TIs solely based on the geometry of the boundary modes then 3D axion insulators are also second-order TIs, as they support surface edge modes, which is not the case.
3. Our protocol for HSCTI is general and it works for any combination of two TIs if the definition “A fusion of two different AZSC topological insulators (TIs) such that they occupy orthogonal Cartesian hyperplanes and their universal massive Dirac Hamiltonian mutually anticommute” is met, see the Abstract. Then we supported this claim with a combination of two strong TIs, topological superconductors, and crystalline insulators in 3D in great details, and in 4D in Appendix E. Please compare with the original paper on HOTI: Science 357, 61 (2017). They wrote only two models, and subsequent works generalized this idea to hundreds of different systems. More than a year after our work appeared on arXiv, our proposal has been generalized in arXiv: 2410.18015 (Ref. 59).
We respectfully disagree with the comment that the topological invariant should be Z not Z2, in support of which the referee quotes our results for crystalline HSCTI with in-plane C4 symmetry from Sec. 4. Consider crystalline QSHI (valley or XY phase) in Table 1 of Nat. Phys. 9, 98 (2013). In the XY phase, the parity eigenvalues at the X and Y points are -1, while at the Gamma and M points they are 1. Thus, their product from all TRIM points is +1. But that does imply that crystalline QSHIs possess a Z invariant. It continues to be a Z2 invariant. It happens generically for crystalline topological phases with Z2 invariant. Exactly, the same situation occurs with the 3D crystalline HSCTI. We clarify this issue in the revised manuscript.
In response to the final “Requested changes” from the referee, we note that in the above discussion we have argued that HSCTI should not be confused with HOTI. Our analysis covers multiple examples in 3D, involving TIs, superconductors, and crystalline TIs, and in 4D to establish the general applicability of the proposed protocol for constructing HSCTI, which has subsequently been extended to encompass other examples in arXiv: 2410.18015 (Ref. 59). It is typically the case that a new general concept is introduced with a few examples, which later gets extended and further generalized, as was the case for HOTIs in Science 357, 61 (2017). Thus, we did not make any claim that is beyond the scope of the general protocol of constructing HSCTI.
Next, we respond to the minor comments raised by the referee.
1. Indeed, in a TRS-odd system TRIM points do not have special meaning. But in lattice-based models there are two components; (a) Hamiltonian which may or may not have TRS, and (b) lattice or crystal which features high-symmetry points in the BZ that occur at the TRIM points. Even for a TRS breaking TI the band inversion occurs at the TRIM points, as is the case for QWZ model for QAHI. We now clarify this issue.
2. The term “hybrid” has already been defined in the Abstract, which we further clarify in the revised manuscript to eliminate any scope of confusion. Also, when a new term is introduced in a scientific article, such as “hybrid” in this case, readers should follow its definition given by the authors. See our responses to Referee 2.
3. The article “the” was used in the first sentence of Sec. 2 carefully, as the model from Eq. (1) captures all the outcomes we discussed in the third paragraph of the Introduction. We now expand this sentence to justify the use of the article “the”.
4. We understand that the referee and the readers would like to know more details about KWANT-based computations. We discussed key theoretical and computational steps in Appendix C. Rest of the details involving Pyhton-based codes will be made available for open access prior to the publication of this manuscript on Zenodo. We hope that the referee will understand that it is very challenging (if possible, at all) to translate Python-based codes in a manuscript.
Changes in the revised manuscript (shown in blue):
1. Right before Eq. (5) and at the end of the second paragraph of Sec. 4, we mention that as HSCTI supports topological modes on the boundaries of a boundary, AZ topology does not apply there, even though its Hamiltonian belongs to one of the ten AZSCs.
2. At the end of the paragraph, following Eq. (5), we argue that HSCTI and axion insulators should not be considered as HOTIs just because they support edge modes on the surfaces. See also `Changes in the revised manuscript’ No. 4 in response to Referee 3.
3. End of first paragraph of Sec. 4: We clarify the topological invariant for crystalline symmetry protected HSCTI by comparing it with a similar phenomenon in crystalline QSHI.
4. At the end of the second paragraph of the Introduction, we state that in TRS breaking systems TRIM points do not have any special importance. But we also say that in lattice-regularized models of such systems, the band inversion still occurs around the TRIM points as they occur at the high symmetry points of the BZ.
5. We expand the first sentence of Sec. 2 to justify the use of the article “the" therein.
Author: Bitan Roy on 2025-04-23 [id 5406]
(in reply to Report 3 on 2024-05-21)We thank the referee for the report. The “Strengths” of the manuscript, “clearly written” and “Could be of interest”, as identified by the referee, are encouraging to the authors.
In response to Weaknesses 1 “Claims made … presented”, we point out that a general principle of constructing HSCTI is proposed in the Abstract and Introduction, which we exemplify with several examples in 3D, including crystalline symmetric systems and superconductors, and in 4D (Appendix E). As such, we could not see such a pedagogical approach as a “Weakness” of a manuscript. For comparison, consider the original paper on higher-order topology (HOT) Science 357, 61 (2017), which outlined a general criterion of realizing HOT by considering only two models (one in 2D and one in 3D). Later, the same principle was invoked to identify a plethora of HOT phases, including superconductors. In this work, we have presented ample examples to establish the general principle of constructing HSCTI in any dimension.
In response to Weakness 2 “Analyses do not … procedure”, we point out that we considered AZSCs for insulators and superconductors, and included phases protected by crystalline symmetries. So, which “state-of-the-art” classification procedure was missed? Different researchers follow complementary approaches for topological classification. It is impossible to commit to all such approaches in one work. We followed the canonical approach based on symmetries, namely the non-spatial time-reversal, particle-hole, and chiral or sublattice symmetries, and subsequently included the crystalline symmetries.
In response to Weakness 3 “Outline of … AZ classes or crystalline invariant”, we were wondering besides AZ and crystalline symmetry classes what other symmetry classes are there, which we might have missed? We are not aware of any other symmetry classification.
In the first paragraph of the “Report” the referee compactly summarized our key results that gave us confidence regarding the clarity of the presentation. Below we respond to the referee’s specific comments and concerns and make necessary changes to the manuscript.
Firstly, we never claimed to invent any “new class”. It is stated in the Abstract while defining the HSCTI “A fusion of two different AZSC topological insulators (TIs) such that they occupy orthogonal Cartesian hyperplanes and their universal massive Dirac Hamiltonian mutually anticommute. The boundaries of HSCTIs can also harbor TIs, typically affiliated with an AZSC different from the parent ones.”, which confirms that the Hamiltonian for HSCTI also belongs to one of the AZSCs. But the AZSC of HSCTI does not support non-trivial AZ topology as supported by examples. Namely, we considered a fusion between class AII 2D QSHI and class BDI 1D SSHI. But the resulting 3D HSCTI belongs to class A that does not have any non-trivial AZ topological invariant. We could not agree more with the referee that any Hamiltonian in any dimension belongs to one of the AZSCs, even when we invoke crystalline symmetries. But that does not imply that every AZSC supports non-trivial topology in every dimension. This is where our construction of HSCTI is novel that allows non-trivial topology beyond the predictions from AZSC. We believe that the referee possibly missed the definition of the HSCTI, which we clarify in the Abstract and Introduction.
HSCTI involves two TIs each coming with its own Wilson-Dirac mass. For such a case our proposed topological invariant from Eq. (5) is operative. In this work, we considered the universal models for QSHI and SSHI which successfully captured all their topological properties, and constructed HSCTI from their hybridization. So, we do not see which aspect of HSCTI is a “model artifact”. Topology can be established from multiple approaches, which include K-theory and homotopy, for example, as the referee pointed out. But we do not have to and cannot pursue all such approaches in one work. If our proposed definition of the topological invariant encounters any shortcoming then the referee should point it out kindly. Any lattice-based Hamiltonian bears some reductant symmetries, such as mirror and rotational symmetries, as is the case for the lattice-regularized BHZ model for QSHI, which does not invalidate the robustness of the topological phases. Furthermore, robustness of our proposed HSCTI has been established by breaking the cubic symmetry and considering the effects of disorder on the quantized surface Hall conductivities. Finally, as a matter of fact, topological invariants of all the AZSCs were computed from the Wilson mass, which later were by K-theory and homotopy, which can also be pursued for HSCTI in the future. However, it is impossible to pursue all approaches in one paper by one research group. As an example, let us count the number of years, research articles, and independent research groups it took to fully understand the ten-fold AZ classification.
We could not appreciate the criticism on the bulk-boundary correspondence, as we computed a bulk-topological invariant and showed that when it is non-trivial there are surface edge modes. Also, we established their robustness against disorder. And we once again stress that a general protocol has been put forward to construct HSCTI in the Abstract and Introduction, which then we exemplified it with a few concrete models. This is exactly how a new concept gets introduced. In this context, we point out that following our work, the proposal for HSCTI has been generalized in arXiv: 2410.18015, going by the name “Boundary topological insulators and superconductors”.
Referee’s “Requested change” for the “title” clearly resulted from the misinterpretation of the word “hybrid symmetry class”, which we have clarified in the rebuttal and revised manuscript. In response to referee’s next “Requested change” for the “general claim” we stress that in this work we have substantiated this claim with multiple examples in 3D and one example in 4D. Also, our proposed construction of HSCTI has now been further explored in arXiv: 2410.18015. Finally, in response to Referee’s last “Requested change” for the “claim of bulk-boundary correspondence”, notice that we proposed a general topological invariant for TIs involving two Wilson-Dirac masses which applies to any Hamiltonian for HSCTI and showed that when it is non-trivial the system supports robust boundary modes, thus meeting the canonical definition of the bulk-boundary correspondence. We will happily revise this discussion if the referee points out an explicit shortcoming of this correspondence. In brief, the topology of a gapped state and its bulk-boundary correspondence cannot be destroyed by infinitesimal perturbations, unless its symmetry is compromised, as shown in by considering the effects of disorder and breaking cubic symmetry.
Changes in the revised manuscript (shown in blue):
1. In the Abstract and Introduction, we now properly define the mathematical definition of “hybridization". In addition, we also explicitly mention the AZSCs of two parent TIs and the resulting HSCTI therein, so that there is no confusion with the fact that HSCTI also belongs to one of the AZSCs in Sec. 2 as well, which, however, does not feature known AZ topology, thereby supporting the novelty of the construction of HSCTI and its topology.
2. At the end of the last paragraph of “Discussions and outlooks" section, we point out that recently our proposed protocol of constructing HSCTI has been generalized in arXiv: 2410.18015 (a new Ref.~60). Along with this preprint, multiple examples we have presented in this work should anchor the general applicability of our proposed protocol.

---

## Round 1 · Referee Report · Anonymous (Referee 2) · 2024-5-18

Strengths
- The authors go in great detail to describe the models at hand.
- It is nice to see that the effect of disorder has been considered in their transport calculations
Weaknesses
-
The manuscript is at times not very clearly written. The topological invariant is discussed quite succinctly. In section 3, they refer to the M and XY phases, which never seem to be define them in the main text (unless I missed them). The authors promise in section 3 they will define them in section 4, but they are only to be found in Appendix D.
-
Other concerns regarding the content are described in the requested changes.
Report
The work and calculations within it seem sound. The concept is interesting. However, to be able to give a recommendation for this paper it would be helpful to have the authors clarify certain points concerning the differences
with existing work. I detail these points and others that I would like the authors to address below.
Requested changes
-
I would like the authors to discuss the similarities and differences with the related concept of Embedded topological insulators, see "Embedded topological insulators" by Tuegel et al. Phys. Rev. B 100, 115126 (2019). At first I was worried that this was the same construction, but here the authors seem to mix symmetry classes, while Embedded topological insulators mix topological insulators of the same symmetry class. A discussion that acknowledges the differences with the idea in this work seems relevant.
-
After reading the manuscript, it is still quite unclear to me why this state is not like a Ferromagnetic, inversion breaking, axion insulator. The authors say that axion insulators display a non-quantized surface conductivity. However I do not think this is generically true. Their comment could refer to 3D TIs that have gapped top and bottom surface states, with half-quantized surface conductivity, like in Ref [33], in which case I believe their comment applies. However, more generally, an axion insulator that breaks T and I could have an integer surface Hall conductivity, if all surfaces are gapped but their gaps have a relative sign difference. Consider as an example a TI without inversion where a bulk Zeeman is chosen to have opposite signs on the side surfaces compared to the bottom surfaces. A construction like this appears for example in Varnava and Vanderbilt Phys. Rev. B 98, 245117 (2018), Fig. 9c) . In this example having the side colors "blue" while the top being "red" of a cubic system seems to realize what is shown in the present paper in Fig. 1aiii . How is this different?
-
I am confused by the bulk invariant, and its relation to other known invariants. First, in-line with comment 2, I think the three-dimensional bulk invariant should be related to the axion angle. Second, if I understood correctly, the invariant is counting two inversions, that of the QSH and that of the SSH. This resembles a second-order topological invariant, where two inversions are required to define the state. Both points are not mutually exclusive, since I beleive axion insulators can be thought of as T- or I-broken HOTIs.
Minor changes:
a. How exactly is the localization of the edge states computed? (red-blue color in Figs. 3 and 9). The authors should be more precise.
b. For the disorder calculations, the authors average over disorder realizations. This defines an error bar to the calculation, the standard deviation, that they could add to Fig. 8.
Recommendation
Ask for major revision
We thank the referee for the report. The referee finds our detailed study and the disorder effects as the “Strengths”, which gives authors confidence regarding the depth of our study. One of the “Weaknesses”, namely “The manuscript … not clearly written” contradicts the comment from two other referees, who praised the clarity of the presentation. In response to referee’s comment in this context “The topological invariant is discussed quite succinctly”, we point out that we discussed the topological invariant in necessary details (as per required) after introducing the model and its phase diagram, as it should be. If the referee has any specific question about the topological invariant, we request him/her to kindly specify it. We will happily address it and expand the discussion. The M phase is defined in the second last paragraph right before Sec. 3. And the XY phase is mentioned at the beginning of Sec. 3 after introducing the corresponding d-vector. Namely, in the XY phase the band-inversion occurs at the X and Y points of the BZ simultaneously that are connected by 4-fold rotations, which we now state clearly at the beginning of Sec. 3. Below we respond to the “Other concerns” by the referee and make appropriate changes to the manuscript.
In the first paragraph of the report the referee compactly summarized our key results, giving us confidence regarding the clarity of the presentation. Despite considering the referee’s suggestion to compare our work with the existing ones as a constructive criticism (which we do nonetheless), we want to point out that this is not a review article and to the best of our knowledge there is not a single model Hamiltonian for 3D systems that on the surface produces “integer” quantum Hall effect. Thus, the novelty of this work should stand on its own ground.
1. In PRB 100, 115126 (2019) authors stack layers of 2D Chern insulator in the z direction, see Eq. (10), which are imbedded within slabs of trivial insulators, see Fig. 10. However, they do not take Chern insulators in different plane and make the corresponding Hamiltonian to anticommute, which is the definition of “hybridization”, a term that we introduced. We cite this paper as Ref. 36 and add a comment on it to contrast this work with ours.
2. While mentioning axion insulators, we should have said that it can also feature a surface “half-quantized” quantum Hall effect, which we now do. Notice that ferromagnetism lifts the Kramer’s degeneracy of bands which is not the case here and the quantized Hall conductivity does not stem from any surface engineering, it is a hallmark of the bulk HSCTI. Also, “quantized” and “half-quantized” Hall effects are two distinct phenomena that shall not be compared. Notice that pi (zero) axion angles can only give half-quantized (trivial) Hall conductivity, besides non-quantized values, but not an integer-quantized Hall conductivity. See Fig. 8 of PRB 98, 245117 (2018), showing that surface Hall conductivity of axion insulators can be either half-quantized or non-quantized. Finally, we never claimed that only HSCTI features surface-quantized Hall conductivity. But to the best of our knowledge there exists no other model in the literature, showing such a phenomenon.
3. As pointed out in the last paragraph that axion angle cannot give integer-quantized surface Hall conductivity. Hence, such a phenomenon for HSCTI is not captured by an axion angle.
The referee correctly pointed out that HSCTI involves inversion of two Wilson-Dirac masses. Although this is the case for HOTIs, the inversion of two Wilson-Dirac mass does not necessarily imply HOT. As such, our model Hamiltonian for HSCTI does not support any quadrupolar or octupolar moment, signature topological invariants of HOTIs.
Next, we respond to the “Minor changes” suggested by the referee.
1. We are confused by the question from the referee “How exactly is the localization of the edge states computed? (red-blue color in Figs. 3 and 9)”. We numerically find the eigenvalues and eigenvectors, and from the spatial profile of the eigenvectors we identify its localization. Prior to the publication of this article, we will make all our codes and results available for open access via Zenodo, as we did for multiple papers in the past.
2. In all numerical transport calculations in disordered systems, we perform disorder averaging, and all the quantities have error bars as mentioned in the captions of Figs. 7 and 8. In the weak and moderate disorder regimes such error bars are extremely small (thus a bit hard to see). In the strong disorder regime, they are more visible. See Fig. 7(c) and Fig. 8 (a).
Changes in the revised manuscript (shown in blue):
1. At the beginning of Sec. 4, we show the d-vector for crystalline TIs as a new Eq. (6) [previous in-text expression]. Immediately after Eq. (6), we define the XY phase.
2. At the end of the third paragraph of Introduction, we contrast our HSCTI with `Embedded topological insulators’ and cite PRB 100, 115126 (2019) as Ref. 37.
3. We now cite PRB 98, 245117 (2018) as Ref. 36, and toward the end of the third paragraph of Introduction we add a discussion on the axion angle induced surface Hall conductivity to argue that it can only give rise to half-quantized or non-quantized surface Hall conductivity, but not to integer-quantized Hall conductivity on the surfaces.
4. At the end of the paragraph after Eq. (5), we state that HSCTI does not possess quadrupole or octupole moments, hallmarks of HOT, and cite Refs. 47-49 where they were first computed.
5. In the captions of Figs. 3 and 9, we specify how to compute the localization of each mode.

---

## Round 1 · Referee Report · Anonymous (Referee 3) · 2024-5-21

(Invited Report)- Cite as: Anonymous, Report on arXiv:scipost_202403_00040v1, delivered 2024-05-21, doi: 10.21468/SciPost.Report.9101
Strengths
- Clearly written
- Could be of interest
Weaknesses
- Claims made as with respect to results presented
- Analyses do not reflect current state-of-the-art understanding of classification procedures
- Outline of conditions, i.e. are the authors considering AZ classes or crystalline invariants
Report
The manuscript reads easily. However, one should be concerned with some of the scientific statements.
First of all the claim of inventing a “new class” is severely out of place. The tenfoldway can be connected to K-theory and is exhaustive provided one asks the right question. That is, once one only considers particle hole symmetry, time reversal and chiral symmetry all possibilities are given by the ten fold way. Here the authors take two orthogonal planes for the two parts of the hybrid Hamiltonian and hence already assume translational symmetry [this is crucial and for example induces weak invariants in a 3D class AII TI]. On top of that they later consider parity and C_4 symmetry. From the perspective of the AZ classification one adds two terms A and B the symmetry of both classes together will determine the AZ classification and under those conditions this is exhaustive, i.e. there is no room for a new class. If one takes into account crystalline symmetries this can be enhanced but the 3D phase still will exhaustively belong to an AZ class with respect to those symmetries. As such the title and some statements are misleading and need to be updated.
Secondly, the authors claim generality but only present a very simple model. Indeed, both parts of the Hamiltonian have many redundant symmetries due to their simplicity [such as a mirror and rotational symmetries]. Such models are fine to illustrate the physics, but if one claims general results a general topological invariant should be presented. No such invariants are given other than the WIlson mass. The latter is fine to tune models but is not related to the possible topological indices without extra steps. If this is a hybrid class in what sense? There are basically only three options: Either they are stable phases and can be captured by K-theory or they can be captured by homotopy theory or they are model artifacts. In this regard I also note the improvised construction although by now several strategies can be used to generally evaluate crystalline TIs.
The same criticism holds for the claimed bulk-boundary correspondence. As before I think a correspondence can only be claimed if there is a direct invariant and a relation to edge does, e.g. as with anomaly inflow arguments for Chern insulators. Apart from a specific model calculator no correspondence in general terms can be concluded from the arguments in the paper. With regard to the stability of the modes [i.e HOTI] discussion is is clear that a symmetry analysis can resolve this as these will not be intrinsic [i.e. due to the SSH part they are more like obstructed TI edge modes].
As such this paper requires drastic changes before publication can be recommended.
Requested changes
- title
- general claims
- claim of bulk boundary correspondence
Recommendation
Ask for major revision

---

## Round 2 · Referee Report · Jasper van Wezel (Referee 1) · 2025-5-9

Report

I thank the authors for considering the points raised in my first report.

The additional clarification of the author's definition of "hybridization" and the added discussion in several places of AZ classes adequately resolve the original ambiguity around how HSCTI fit into the AZ classification.

The added definitions and discussion of HOTI and axion angle now also make sufficiently clear what the authors consider to be essential differences between them and HSCTI.

I would personally still prefer a slightly more careful phrasing of claims about the generality of the proposed HSCTI construction and their properties, but I appreciate the replies of the authors and don't think this should affect publication.

I believe the authors adequately addressed all issues raised, and I recommend publication.

Recommendation

Publish (easily meets expectations and criteria for this Journal; among top 50%)

---

## Round 2 · Referee Report · Anonymous (Referee 2) · 2025-5-14

Report

I thank the authors for their replies, which have addressed my comments.

I still have thoughts about the axion angle point, becauset I believe it is possible to write an axion field theory that leads to an integer Hall effect at the boundary . What is the correct field theory of this hybrid phase is, in my opinion, an open question, which I do not think should prevent publication since this is not the goal of the paper. I am thus happy with the additional comments that point out the differences between current versions of axion insulators.

To clarify one of my previous comment, localization length can be calculated in equivalent, but not numerically equal, ways (e.g. IPR or by fitting the real-space wave-function profile, as they seem to do). I am happy that they now clarify how this is done precisely and that they will share their codes.

Recommendation

Publish (easily meets expectations and criteria for this Journal; among top 50%)

---

## Round 2 · List of Changes

Changes in the revised manuscript in response to Referee 2 (shown in blue):

1. In the Abstract and Introduction, we now properly define the mathematical definition of `hybridization’. In addition, we also explicitly mention the AZSCs of two parent TIs and the resulting HSCTI therein, so that there is no confusion with the fact that HSCTI also belongs to one of the AZSCs in Sec. 2 as well, which, however, does not feature known AZ topology, thereby supporting the novelty of the construction of HSCTI and its topology.

2. At the end of the last paragraph of `Discussions and outlooks’ section, we point out that recently our proposed protocol of constructing HSCTI has been generalized in arXiv: 2410.18015 (a new Ref.~60). Along with this preprint, multiple examples we have presented in this work should anchor the general applicability of our proposed protocol.

Changes in the revised manuscript in response to Referee 3 (shown in blue):

1. At the beginning of Sec. 4, we show the d-vector for crystalline TIs as a new Eq. (6) [previous in-text expression]. Immediately after Eq. (6), we define the XY phase.

2. At the end of the third paragraph of Introduction, we contrast our HSCTI with `Embedded topological insulators’ and cite PRB 100, 115126 (2019) as Ref. 37.

3. We now cite PRB 98, 245117 (2018) as Ref. 36, and toward the end of the third paragraph of Introduction we add a discussion on the axion angle induced surface Hall conductivity to argue that it can only give rise to half-quantized or non-quantized surface Hall conductivity, but not to integer-quantized Hall conductivity on the surfaces.

4. At the end of the paragraph after Eq. (5), we state that HSCTI does not possess quadrupole or octupole moments, hallmarks of HOT, and cite Refs. 47-49 where they were first computed.

5. In the captions of Figs. 3 and 9, we specify how to compute the localization of each mode.

Changes in the revised manuscript in response to Referee 1 (shown in blue):

1. Right before Eq. (5) and at the end of the second paragraph of Sec. 4, we mention that as HSCTI supports topological modes on the boundaries of a boundary, AZ topology does not apply there, even though its Hamiltonian belongs to one of the ten AZSCs.

2. At the end of the paragraph, following Eq. (5), we argue that HSCTI and axion insulators should not be considered as HOTIs just because they support edge modes on the surfaces. See also `Changes in the revised manuscript’ No. 4 in response to Referee 3.

3. End of first paragraph of Sec. 4: We clarify the topological invariant for crystalline symmetry protected HSCTI by comparing it with a similar phenomenon in crystalline QSHI.

4. At the end of the second paragraph of the Introduction, we state that in TRS breaking systems TRIM points do not have any special importance. But we also say that in lattice-regularized models of such systems, the band inversion still occurs around the TRIM points as they occur at the high symmetry points of the BZ.

5. We expand the first sentence of Sec. 2 to justify the use of the article `the’ therein.

Unsolicited changes in the revised manuscript (NOT shown in blue):

1. We expand the Abstract to include more important details of our study.

2. We display the roadmap of the paper as a separate subsection, Sec. 1.2 and expand it by including details of the materials covered in various appendices.

3. In addition, we made multiple cosmetic changes to improve overall lucidity of the presentation without altering any major or minor conclusions.

---

## Editorial Decision

accepted_in_target_journal